# Central amygdala single-nucleus atlas reveals chromatin and gene transcription dynamics in human alcohol use disorder

Che Yu Lee[1,7], Ahyeon Hwang[1,2,7], Delaney McRiley[3,7], Jaywon Lee[3], Genevieve Thibodeau[3], Catharine Duman[3], Xiangyu Zhang[4], Mario Skarica[3], Jensine Coudriet[3], Siwei Xu[1], Rosemarie Terwilliger[3], Alexa-Nicole Sliby[3], Jiawei Wang[3], Tuan Nguyen[3], Yujing Liu[3], Hongyu Li[4], Yi Dai[1], Ziheng Duan[1], Yutong Lei[1], Yingxin Lin[4], Jill R. Glausier[5], David A. Lewis[5], Joel Gelernter[3], Paul E. Holtzheimer[6], Ke Xu[3,6], Hang Zhou[4], Hongyu Zhao[4], Summer L. Thompson[3], John H. Krystal[3], Alicia Che[3], Jane R. Taylor[3], Jing Zhang[1,2] ✉ & Matthew J. Girgenti[3] ✉

Regulation of gene expression is a highly coordinated process in both the healthy and pathological brain with unique patterns across a multitude of cell types. Here we present a multi-omic single nucleus study of ~175,000 nuclei from 50 donors with alcohol use disorder (AUD) and control donors without AUD, profiling cell type specific gene expression and chromatin accessibility in the human central amygdala. We identify all major CNS cell types and neuronal subtypes and find inhibitory neurons are particularly affected by AUD. We find high numbers of differentially expressed genes (DEGs) including *GABRA2*, *GRM8*, and *NCAM1* and show significant enrichment for AUD risk genes within these DEGs. We identified 51,431 cell type-specific, disease associated candidate *cis*-regulatory elements including an interneuron-associated set of chromatin loops at the AUD risk gene *CALN1*. Transcription factor footprinting identified Kruppel-like factors upstream of AUD GWAS genes and DEGs. Finally, we also perform cell type-specific fine mapping for AUD GWAS to prioritize variants within functional genomic elements.

Alcohol use disorder is a common neuropsychiatric disorder that is accompanied by significant social and vocational dysfunction, numerous medical co-morbidities and increased mortality. AUD, including alcohol abuse and dependence, is moderately heritable and is caused by genetic and environmental factors[1,2]. AUD has a prevalence of about 6.2% in the adult American population and genetic studies suggest heritability of 50–60%[3–6].

Genome-wide association studies (GWAS) have demonstrated that genetic variation also contributes to individual vulnerability to AUD[7]. There are many published AUD GWAS that have identified genome wide-significant risk loci[1,5,8–17]. However, the link between genetic risk and molecular brain pathology is still far from clear. The pursuit of the molecular underpinnings of AUD risk is greatly complicated by the direct and indirect consequences of acute and chronic

[1]Department of Computer Science, University of California, Irvine, CA, USA. [2]Mathematical, Computational and Systems Biology Program, University of California, Irvine, CA, USA. [3]Department of Psychiatry, Yale University School of Medicine, New Haven, CT 06520, USA. [4]Department of Biostatistics, Yale University School of Public Health, New Haven, CT 06510, USA. [5]Department of Psychiatry, University of Pittsburgh School of Medicine, Pittsburgh, PA 15213, USA. [6]Department of Psychiatry, Geisel School of Medicine at Dartmouth, Lebanon, NH 03756, USA. [7]These authors contributed equally: Che Yu Lee, Ahyeon Hwang, Delaney McRiley. ✉e-mail: zhang.jing@uci.edu; matthew.girgenti@yale.edu

alcohol exposure and other AUD-related effects (e.g. malnutrition)[18]. Current clinical and basic research suggests important, complex, and currently inadequately understood interactions between alcohol use and AUD neuropathology[19,20]. Alcohol crosses the blood brain barrier and triggers changes in the CNS, including reduced glucose uptake and disturbances in neurotransmitter release including glutamatergic, GABAergic and dopaminergic alterations. Although the mechanistic basis for these interactions is currently not well understood, candidates from previous studies have included alcohol effects on neurons and astrocytes[21,22], inflammatory cytokines and immune mediators[23], brain inhibitory function[24,25] and synaptic signaling[26] specifically at $GABA_A$ receptors, where ethanol can directly bind and modulate behaviors including anxiolysis, disinhibition, and hyperlocomotion. This suggests alcohol may alter the epigenome resulting in long lasting aberrant effects on the brain related to these processes[13].

Single-cell atlases have emerged as a promising tool for determining both the gene regulatory mechanisms and functions of disease related genes in specific brain regions. Consistent with preclinical studies, neuroimaging studies in AUD patients implicate multiple brain regions in impulsivity, alcohol reward, and craving including the medial prefrontal cortex (PFC)[27], nucleus accumbens[28], amygdala[29] and hippocampus[30]. Recent studies have demonstrated alcohol-use disorders are associated with negative reinforcing states and that individuals may drink alcohol to reduce anxiety or withdrawal symptoms[31]. A subset of individuals self-medicate with alcohol to reduce anxiety and stress and these stress-dampening properties are mediated through the amygdala[32]. The central nucleus of the amygdala (CeA) is the primary output nucleus of the amygdala and is largely made up of inhibitory neurons. It has also been implicated in the reinforcing effects of alcohol and other drugs of abuse[33]. To date there have been no single-cell multi-omic studies of the human CeA in AUD. Therefore, we performed single nucleus (sn)ATAC-seq and snRNA-seq from the same nuclei (snMultiome) to define AUD-associated gene regulatory programs at the epigenomic and transcriptomic levels. This work provides a powerful look into the cellular heterogeneity of the CeA and allows us to unravel critical biological systems underlying alcohol dependence in the amygdala.

Here we present a single cell-type multi-omic analysis of 174,188 high-quality nuclei (average 6028 reads/nuclei) from postmortem human brain tissue of AUD and control donors without AUD (CON; $N = 50$). We directly integrated snRNA and snATAC data from the same nuclei, providing us with the most complete understanding of gene regulatory systems affected by AUD. We provide a cell type specific molecular atlas of the CeA and identified all major CNS cell types including six neuronal subtypes across inhibitory neurons. We identified >1800 cell type-specific AUD DEGs which were primarily within inhibitory and excitatory neurons and astrocytes. We also found significant enrichment of AUD risk genes within our neuronal subtypes. Analysis of chromatin assemblies in the CeA allowed us to identify candidate cis-regulatory elements and disease-associated changes in transcription. We identified transcription factors that were upstream of AUD risk genes and cell type-specific DEGs using gene regulatory network machine learning. We fine-mapped GWAS signals at select Problematic Alcohol Use (PAU; a trait that combines AUD and alcohol related problems) risk loci with our chromatin accessibility data to link AUD risk signals to specific cell types and to nominate candidate lead SNPs. This functional genomic approach allowed us to both confirm GWAS fine mapping from previous studies (CACNA1E, ADH1B, ADH1C, GABRA4) and to identify new lead SNPs for other AUD genes (SEMA6D, PPP1R13B, CACNA1C, NF1). Taken together, we present an enormous resource integrating AUD genetic risk with single cell-type gene regulatory information of the AUD central amygdala, highlighting the vulnerability of inhibitory neurons in AUD pathophysiology and identifying several candidate genes for future study.

## Results

### Single cell genomic atlas of AUD central amygdala

We used single-nucleus multiome-sequencing (snMultiome-seq) to profile nuclei isolated from the central amygdala of postmortem brain of 50 donors from University of Pittsburgh Medical Center brain bank. The sample included 22 donors with a history of alcohol abuse and dependence (alcohol use disorder) and 28 CONs with no history of alcohol abuse. Complete demographics for the entire cohort are included in Supplementary Data 1. To dissect the single cell-type gene expression and gene regulatory changes in the AUD brain, we used a custom bioinformatic pipeline for analysis of each genomic layer and integration[34]. After rigorous quality-control and filtering, we report results from a discovery cohort of 174,188 droplet based nuclei (2081 genes and 6028 transcripts per nuclei) from snMultiome (10X Chromium Single Cell Multiome ATAC + Gene Expression) (Supplementary Figs. 1, 2)[35]. The snMultiome technology uniquely tags DNA and RNA from the same cell to measure chromatin accessibility and transcript levels from the same nuclei.

After UMAP clustering, marker genes analysis and comparison with previous studies of the whole amygdala, we annotated all major cell types in the CNS including inhibitory neurons (INH; 23,876 nuclei, 13.71%), excitatory neurons (EXC; 8643 nuclei, 4.96%), oligodendrocytes (OLI; 80,549 nuclei, 46.24%), oligodendrocyte progenitor cells (OPC; 15,341 nuclei, 8.81%), endothelial cells (END; 3316 nuclei, 1.90%), astrocytes (AST; 27,313 nuclei, 15.68%) and microglia (MIC; 15,150 nuclei, 8.70%). Notably, INH neurons represent ~75% of the central amygdala neurons (with a median of ~80% INH neuron composition across all samples) consistent with previous neuroanatomical counts[36]. To visualize our data across genomic modalities, we generated UMAP clusters for each using the same nuclei. RNA and ATAC UMAPs were created based on gene expression levels (Fig. 1a) and chromatin accessibility (Fig. 1b), respectively. Gene expression and chromatin accessibility from the same nuclei were integrated to generate a snMutliome UMAP (Fig. 1c). We calculated cell type proportions based on the number of nuclei isolated (Fig. 1d) and found no significant proportion shifts due to disease (AUD versus CON): INH P-value = 0.718; EXC P-value = 0.324; OLI P-value = 0.977; OPC P-value = 0.406; END P-value = 0.551; AST P-value = 0.945; and MIC P-value = 0.945. We plotted average gene expression levels for each cell type marker transcripts (Fig. 1e), and normalized chromatin signals for open ATAC peaks for each marker gene (Fig. 1f). Transcriptomic subtypes were largely conserved across individuals (Supplementary Fig. 2a,b).

With the aid of previously observed cell type markers of the amygdala[36-38] and using de novo marker detection, we were able to identify subtypes for both our neuronal and non-neuronal cells. We identified six INH neuron subtypes including DRD1, LHX9, SST, VIP, PENK, and FREM1 positive, two oligodendroctye subtypes, differentiated endothelial and mural cells, and separated microglia and macrophages, with numbers of cells significantly higher than what has been previously reported[38] (Fig. 2a). Average gene expression levels for individual marker transcripts were used to classify each subtype (Fig. 2b). We found the highest percentage of INH neurons were the DRD1 (31.6%) and LHX9 (21.7%) positive neurons. The percentage of the remaining INH subtypes were SST, 12.8%; VIP, 9.7%; PENK, 14.2%; FREM1, 10.0% (Fig. 2c). We identified two OLI subtypes of unequal proportion proportions including oligo-1, 78.2%; oligo-2, 21.8%) (Fig. 2d). We identified two vascular subtypes (endothelial cells, 39.8% and mural cells 60.2%) (Fig. 2e); and two immune subtypes (microglia, 97.7% and macrophages, 2.3%) (Fig. 2f). To confirm our immune cell subtypes, we examined expression of additional microglia-specific markers P2RY12, TMEM119, and found high co-expression with APBB1IP. Additionally, we examined the expression of infiltrating macrophage-specific marker SIGLEC1 (aka CD169) and found high co-expression with our F13A1 macrophage cluster (Supplementary Fig. 3a). Cell subtypes were largely conserved across individuals, as most of the identified cellular

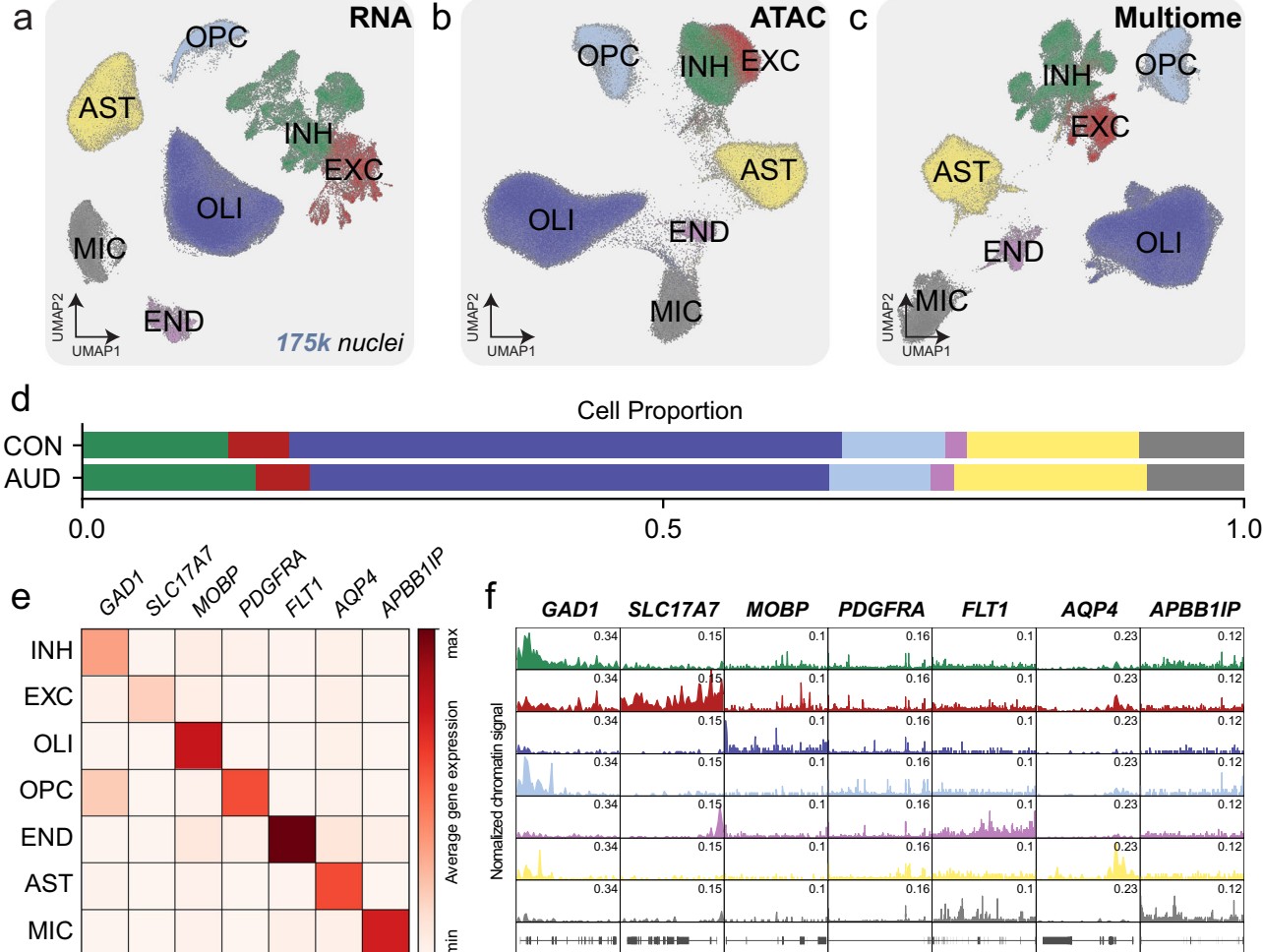

**Fig. 1 | Multimodal genomic taxonomy of cell types in the human central amygdala. a** UMAP visualization of snRNA-seq across seven cell types. **b** UMAP visualization of snATAC-seq across seven cell types. **c** UMAP visualization of snMultiome across seven cell types. **d** Cell proportion of cell types across diagnostic conditions (CON, AUD). **e** Average single nucleus gene expression heatmap of canonical cell type marker genes across seven cell types in snRNA-seq. **f** Normalized chromatin accessibility profiles of canonical cell type marker genes across seven cell types in snATAC-seq.

subsets contained nuclei from the majority of donors (Supplementary Fig. 2a). To confirm selected interneuron markers, we used immunohistochemistry to colabel each marker with the GABAergic marker *GAD1*. We used acetylcholinesterase histochemistry to confirm the location of the CeA (lightest stain) compared with surrounding amygdala structures (Supplementary Fig. 3b). We confirmed co-expression of PENK (Supplementary Fig. 3c) and FREM (Supplementary Fig. 3d) with GAD1 in amygdala tissue sections adjacent to those used for acetylcholinesterase staining.

### Systematic differential gene expression implicates inhibitory neuron vulnerability in AUD

To better determine how individual cell types are affected by alcohol abuse, we analyzed differential gene expression across all cell type clusters. To ensure the highest rigor in our differential gene expression analysis, we combined two extensively benchmarked methods for DEG detection: parametric-based MAST[39] and permutation-based Wilcoxon Test[40] (Methods). For each cell cluster, we report all DEGs that were identified as overlapping between the two pipelines (Supplementary Data 2). For our DEG analysis, we analyzed canonical cell types (Fig. 3a, left panel) but also examined gene expression in specific INH neuronal subtypes (Fig. 3a, right panel). DEGs significantly associated with AUD were identified in all cell types. We also found that DEGs (769) were enriched in INH neurons compared to the EXC neurons, with the

majority (466, 60.6%) occurring in INH *PENK* neurons. We identified 4364 unique protein-coding AUD DEGs using MAST (Supplementary Data 3) and 3290 unique protein-coding DEGs from the Wilcoxon test (Supplementary Data 4). Of those, 2270 DEGs overlapped in the same direction, shared directional fold change (FC) > 1.2 with FDR < 0.01, and included 1805 unique genes (Supplementary Data 5). In general, we found that neuronal cell types shared the highest gene expression changes, with 5.93% DEGs shared between both EXC and INH classes (Fig. 3b). No gene was regulated across all 7 major cell types. Using cosine similarity, we compared AUD DEGs across cell types and found the highest concordance of gene expression among neuronal subtypes (Supplementary Figs. 4a, 3a). We also performed DEG analysis on all cell subtypes (Supplementary Fig. 5b,c). In general, we found that the subtypes shared the majority of DEGs with their major cell type clusters (~89% of up regulated and ~98% of down regulated DEGs) (Supplementary Fig. 5a) with 95% of DEGs identified in subtypes also found in the major cell types (Supplementary Fig. 5d).

To assess the impact of drug use and comorbidities, we performed additional analyses that included covariates for manner of death, lifetime alcohol consumption, illicit drug use, comorbid psychiatric disorders, and medications at time of death (Supplementary Data 6). Across cell types, ~80% of DEGs overlapped with our model (Supplementary Fig. 6a). Additionally, to ensure rigor, we tested a sample-based DEG method (i.e. pseudobulk) using DESeq2[41]

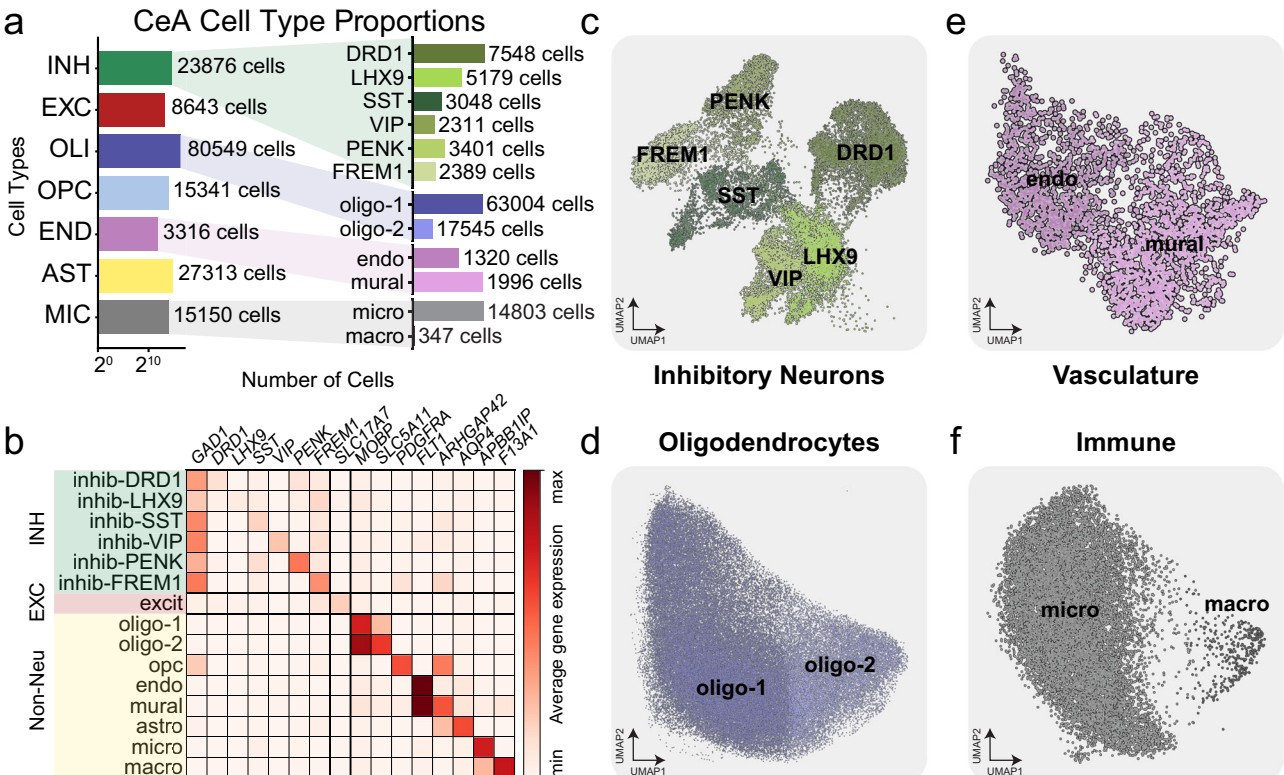

**Fig. 2 | Transcriptomic atlas of cell subtypes in the human central amygdala.** **a** Bar plots of each of the seven canonical cell types numbers (log2 scaled) and the magnified proportion of cell subtypes within INH, OLI, END, and MIC, respectively. **b** Average single nucleus gene expression heatmap of both cell type and cell sutype marker genes across fifteen cell subtypes in snRNA-seq. **c** UMAP visualizatwion of snRNA-seq across the six INH subtypes. **d** Two OLI subtypes. **e** Two vasculature (END) subtypes. **f** UMAP and two immune (MIC) subtypes.

(Supplementary Data 7). We identified additional DEGs: 14 in inhibitory neurons, 26 in excitatory neurons, 1 in oligodendrocyte, 10 in astrocytes, and 1 in microglia (Supplementary Fig. 6b). Finally, we also observed sex-specific responses (Supplementary Data 8–10), consistent with previous observations in AUD[42,43]. Specifically, males have the most DEGs in the INH cell type, whereas female DEGs are more uniformly distributed across cell types (Supplementary Fig. 6c, e). We also found pathway enrichment differences, specifically in WNT5A signaling in females and ERBB4 signaling in males (Supplementary Fig. 6d, f). In addition, male pathways tended to be more cell type-specific, while females' tended to be enriched across cell types. 710 DEGs were common across the male and female specific comparisons or approximately 55% or 25% overlap, respectively.

Among top DEGs per major cell type, we found little overlap (Fig. 3c). Due to the high sensitivity of our analyses, we were able to identify several genes previously implicated in alcohol addiction including *KCNIP1* in both INH neurons and MIC, as well as *VAMP2* which has previously been shown to stimulate GABA neuro-transmitter release after alcohol intake[44]. To better understand the global molecular processes affected by AUD, we performed gene set enrichment analysis on all 1805 AUD DEGs (Supplementary Fig. 4b and Supplementary Data 11). We observed significant enrichment for neuronal pathways related to axon guidance, glutamate receptors and cell to cell signaling. Individual cell type-specific gene set enrichment analysis (Fig. 3d, Supplementary Fig. 4c, Supplementary Data 12, 13) identified pathways related to GABA synthesis, glutamate receptor binding, and P2Y receptor signaling. Taken together, differential gene expression changes in AUD amygdala are mostly neuronal and affect systems related to neurotransmitter signaling and immune processes.

Interestingly, we observe a statistically significant enrichment of 19 PAU/AUD GWAS risk genes (*P*-val=5.32e−3, O.R. = 2.23) (Fig. 3e and Supplementary Fig. 7a,b). We highlight the gene expression changes in the INH neurons of the amygdala (Fig. 3e) as these are the most abundant neuronal cell type and have the most number of DEGs. We found significant decreases in *GRM8* (logFC = −0.485 and FDR = 5.50e−5) and increases in *GABRA2* (logFC = 0.588 and FDR = 6.73e−19 in DRD interneurons. We also observed an increase in *GRM8* (logFC = 1.26 and FDR = 1.73e−4) and a decrease in *NCAM1* (logFC = −0.353 and FDR = 1.06e−12) in PENK neurons and we confirmed these expression changes using fluorescent in situ hybridization (Supplementary Fig. 7c-j). There is extensive evidence for a role of glutamate neurotransmission dysfunction in alcohol dependence[45,46] and related addiction behaviors and several metabotropic glutamate receptors have been implicated in these processes[47].

## Cis-gene regulation of calcium activity and glutamatergic signaling genes is disrupted in AUD

In addition to measuring gene expression differences in CeA of the AUD brain, we measured chromatin accessibility within the same nuclei. This allowed us to identify cell type-specific open chromatin peaks and *cis*-gene regulatory mechanisms. Since the majority of common risk variants for neuropsychiatric disorders are located within non-coding regions of the genome, they likely functionally disrupt gene expression through *cis*-regulating elements (CREs). To identify open chromatin peaks we utilized MACS2[48], and found 789,247 peaks across all cell types, the majority of which were located in introns (50.0%) or distal intergenic regions (24.9%) (Fig. 4a, Supplementary Data 14). We then used the merged peak set of fixed-width 501-bp union peaks (378,931 total) based on the iterative overlap peak

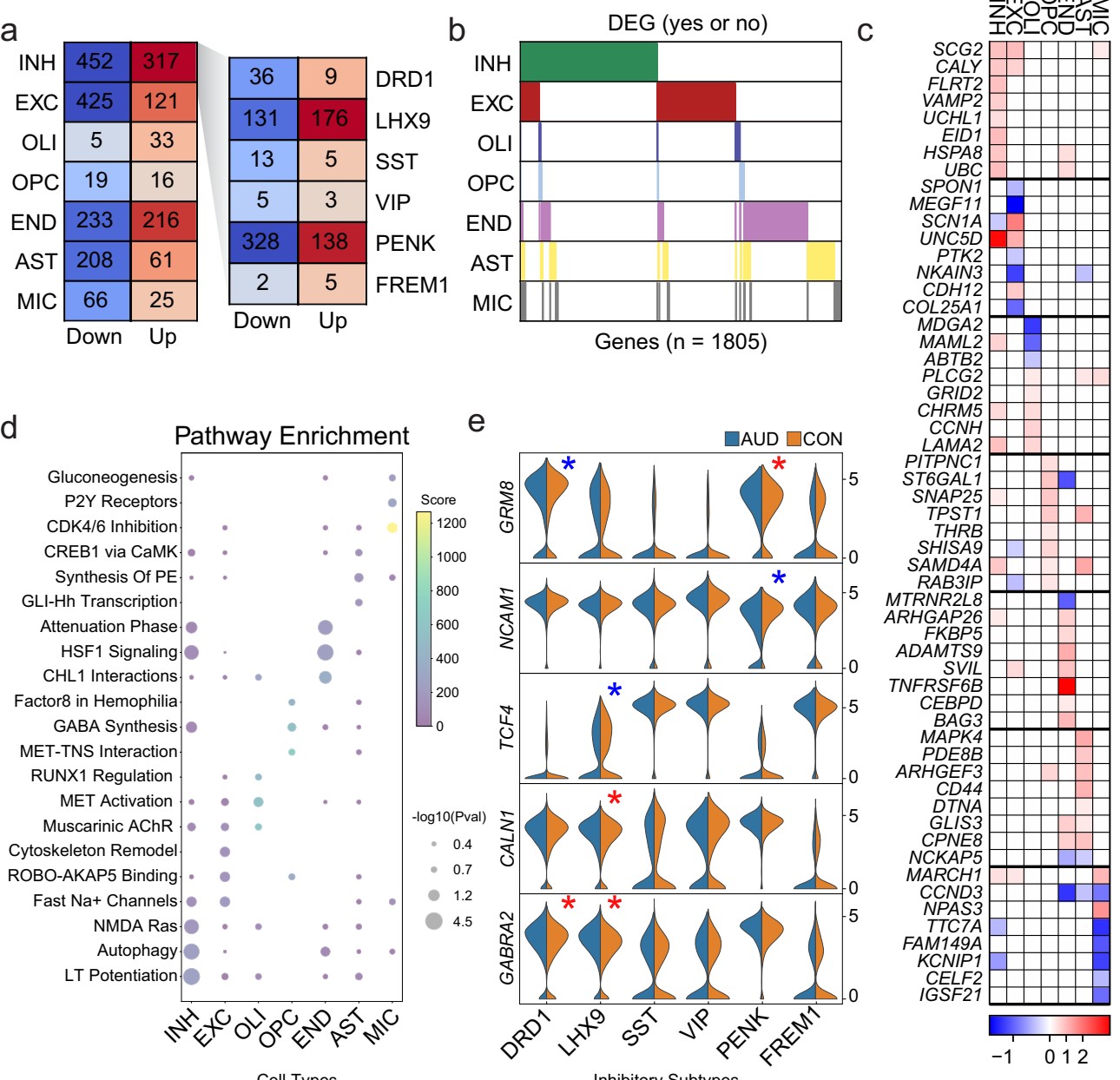

**Fig. 3 | Cell type-specific gene expression changes in AUD pathology.**
**a** Significant DEG counts in both directions for each major cell type (left) and INH sub cell types (right). The counts for INH are determined by taking the unique set of DEGs in their subtypes. DEGs met a threshold of FC > 1.2 and FDR < 0.01 and overlap between MAST and Wilcox analyses. The intensity of colors (blue downregulated and red upregulated) is proportional to the entry values. **b** Binary plot indicating whether a gene (column) is a DEG in a given cell type (row) or not (n = 1805 unique DEGs from the seven cell types in 2a). **c** Heatmap of MAST log2FC values of the top FDR significant DEGs for each cell type. For INH, the MAST log2FC value of the subtype with the most significant FDR is shown. **d** Top enrichR Reactome (Pathway) terms for the DEGs from 2a, (Fisher exact test with corrections for multiple comparisons). **e** Violin plot showing AUD versus CON expression of *GRM8* (FDR = 5.50e−5, 1.73e-4 for *DRD1*, *PENK* interneurons respectively), *NCAM1* (FDR = 1.06e−12), *TCF4* (FDR = 7.99e−8), *CALN1* (FDR = 2.25e−6), *GABRA2* (FDR = 6.73e−19, 4.19e−15 for *DRD1*, *LHX9* interneurons respectively) in each INH subtype with significant up (red asterisk) and down (blue asterisk) expression (one-sided Wald's test with corrections for multiple comparisons).

merging procedure which preserves cell type-specific peak annotation (Methods). This union peakset consists of peaks across all seven canonical cell types and is used in all downstream analyses (Supplementary Data 15). We found that nearly half of all peaks are cell type-specific, highlighting the importance of studying individual cell type epigenomic signatures in the brain (Supplementary Fig. 8a). We found the highest overlap in peaks between EXC and INH neurons (36% Jaccard index overlap). We identified robust peaks for cell type markers for the 7 major cell types (Supplementary Fig. 8b). Additionally, we

identified peaks in cell subtypes and found that a significant portion (55.1%) differed from their major cell types (Supplementary Fig. 8c, d) with most peaks in INH *DRD*+ and *PENK*+ neurons, suggesting a high degree of specificity in gene expression control among INH subtypes. Finally, we performed gene ontology enrichment on cell type marker peaks and found that they were enriched for expected cell type-specific biological processes (Supplementary Fig. 8e),

We then found 1,526,912 peak-to-gene links (correlation>0.45, FDR < 1 × 10⁻⁴) using the union peaks and plotted side by side heatmaps

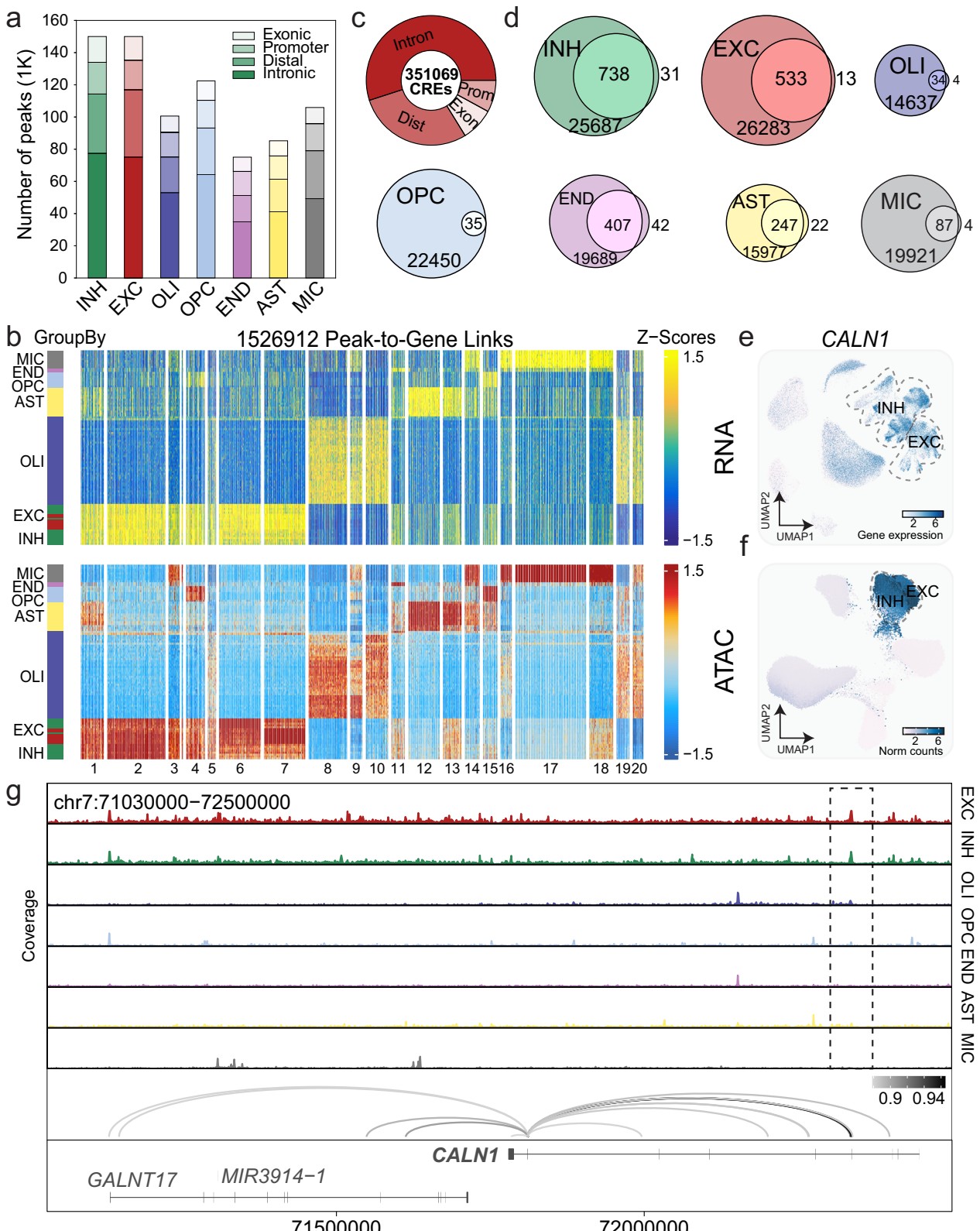

**Fig. 4 | AUD alters *cis*-regulation of gene expression across cell types. a** Stacked barplot of the number of peaks separated by genomic category (Intron, Distal, Promoter, Exon) for the major cell types in paired snATAC-seq. **b** Side by side heatmaps of linked ATAC (top) and gene (bottom) regions. Rows are clustered using *k*-means with k = 20. 1,526,912 peak-to-gene links are shown using thresholds: FDR < 1×10⁻⁴, correlation > 0.45, and Variance Quartiles > 0.25. **c** Donut plot showing the number of CREs (*n* = 351,069) that have peak-to-gene correlation > 0.4,

FDR < 0.05 separated by genomic category. **d** Venn diagrams showing the intersection of CRE-linked-genes and AUD-DEGs for each cell type. UMAP plots of *CALN1* snRNA-seq gene expression on the **e**, snRNA (top) and **f**, snATAC-seq (bottom) embeddings. Neuronal cell types with high expression of *CALN1* are outlined by dashed lines. **g** Chromatin accessibility signal tracks highlighting *CALN1* peak-to-gene links across cell types. Differences across the seven cell types are indicated by the dashed boxes.

of linked ATAC regions and gene expression (Fig. 4b). Peak-to-gene links are correlations between chromatin peak accessibility and nearby gene expression that can be calculated by integrating the modalities of snMultiome data. Since snMultiome simultaneously profiles the gene expression and chromatin accessibility of the same nuclei, our peak-to-gene links profiles are matched from the same nuclei. Previous studies have shown that not all peaks are CREs controlling gene expression in human brain tissue[49]. To determine which peaks are most likely to regulate gene expression, we calculated the correlation between peak accessibility and target gene expression (threshold of correlation > 0.4 and FDR < 0.05). We found that across all cell types approximately 92.64% of peaks showed strong gene expression regulatory activity (Fig. 4c).

Because we identified an extensive number of DEGs across all cell types and significant enrichment of DEGs near risk variants for AUD, we reasoned that *cis*-regulatory architecture may be altered in the AUD brain. To identify disease gene regulatory mechanisms, we intersected our CRE linked genes (CLGs) with DEGs for each cell type. 36.05% of CREs were linked to an upregulated DEG. We found the most overlap in CLGs in INH neurons corresponding to the high number of DEGs we observe in those cells (Fig. 4d). We also found that CLG target genes that overlap between cell types are typically linked to different CREs. For example, *GABRA2* is upregulated in INH neurons and down regulated in AST (Fig. 3e and Supplementary Data 16) with each having unique, non-overlapping CREs. We also highlight the peak-to-gene links for AUD risk gene *CALN1*. *CALN1* is a calcium binding gene that is highly and exclusively expressed in the neuronal cell types (Fig. 4e,f) and was significantly upregulated in INH (Fig. 3e). We find an extensive number of ATAC peak-to-gene links (strong CREs, correlation>0.85) to *CALN1*'s TSS which are all cell type-specific (Fig. 4g dashed box).

We also identified peak-to-gene links (strong CREs, correlation>0.8) for genes previously implicated in addiction in other cell types. For example in astrocytes, we find an extensive number of ATAC peak-to-gene links to *NKAIN3* and *FGFR3*. *NKAIN3* is a sodium/potassium pump gene that is highly expressed in astrocytes (Supplementary Fig. 9a, d) and changes in *FGFR3* transcript have been linked to alcohol exposure of offspring of binge drinking behavior[50] (Supplementary Fig. 9b, e); both TSS's are cell type-specific to astrocytes (Supplementary Fig. 9c, f dashed box). Additionally, both genes were significantly down regulated in AST (Supplementary Fig. 9g). Taken together, these results demonstrate unique epigenomic processes regulating gene expression in CeA and point to mechanisms affecting excitatory and calcium signaling driven by AUD.

## Cell type-specific TF gene regulation alters GWAS and DE genes in AUD

We mapped transcription factor binding motifs within our CREs that correspond to differentially expressed genes in cells from our AUD cohort to understand the upstream regulatory mechanisms governing gene expression. We used chromVAR[51] to compute TF motif variability in individual cell clusters by estimating the enrichment of TF binding motifs in accessible chromatin regions by cell type (Supplementary Fig. 10a). To obtain motifs with high enrichment in the CeA, we prioritize peak-to-gene links connecting to AUD DEGs (Supplementary Fig. 10b). We highlight TF *RFX2* which has high enrichment in the INH and EXC neurons and AST (Fig. 5a). We then performed TF footprinting analysis to investigate binding dynamics of *RFX2* within the genome of these cell types (Fig. 5b) and found significant enrichment for *RFX2* motifs in both EXC and INH neurons.

To calculate disease-specific differences in chromatin accessibility, we focused on peak-to-gene linkages specific for AUD. After calculating cell type-specific peak-to-gene links across all samples, we isolated peaks with strong differential peak-to-gene correlation (*Pearson* Correlation > 0.5) in the AUD cohort and considered these 'AUD-specific peaks' for downstream analysis (Fig. 5c and **Methods**).

We found the majority of AUD-specific peaks in OPCs and END (24.13 and 24.09%, respectively) and INH (21.77%) (Supplementary Data 17–23). We built cell-type-specific gene networks (Supplementary Data 24–30) and INH motifs were chosen based on the number of DEGs and GWAS genes (Supplementary Data 31). We then performed enrichment analysis using these AUD-specific peaks and found an overrepresentation of TFs belonging to the KLF (Kruppel-like) family (Fig. 5d and Supplementary Data 32). We found significant enrichment for KLF16 motifs in INH, END, and MIC cells (Fig. 5e). Transcription factor footprinting of *KLF16* revealed stronger enrichment in AUD INHs compared to CON INHs (Fig. 5f), reflecting the abundance of KLF motifs in the AUD peak-to-gene links motif enrichment described above. We constructed a gene regulatory network for AUD INHs (Fig. 5g) and found *KLF16* was linked to 629 DEGs and 64 AUD risk genes including *GRIN2A* (upregulated), *GRIK1* (upregulated), *SMAD3* (downregulated), and *OPRM1* (risk gene) (Fig. 5h). We found significant enrichment for DEGs in predicted downstream targets of the KLFs (P-val 1.26e−53; O.R. = 4.052) and nominal significance for GWAS genes downstream (P-val 4.052; O.R. = 0.643).

In order to functionally validate our genomic findings, we employed a mouse model of alcohol abuse. We isolated frozen amygdala from mice treated with an 2 g/kg injection of 100% EtOH. We performed single nuclei isolation and snMultiome in the same manner as our human donor cohorts. We identified all major cell types across all modalities: RNA (Supplementary Fig. 11a), ATAC (Supplementary Fig. 11b) and combined Multiome (Supplementary Fig. 11c) for 19,072 individual nuclei. We identified 1814 total DEGs across all major cell types with most falling in the INH (Supplementary Fig. 11d). When compared with the human AUD DEG dataset we found overlap for 678 DEGs (37.6%) (Supplementary Fig. 11e). Most DEGs overlapped in neuronal cell types and this was reelected in gene set enrichment analysis which identified terms related to neuron projection, synaptic transmission, and glutamatergic signaling (Supplementary Fig. 11f). In addition, we intersected our INH DEGs with a previous central amygdala mouse model of alcohol withdrawal[52] and found 38 overlapping DEGs in INH and EXC neurons and MIC (Supplementary Fig. 11g). We performed peak-to-gene linkage analysis on our mouse snMultiome data and identified 7485 total in the mouse AUD model. Of these, 4710 overlapped with the human AUD CRE genes (62.9%) (Supplementary Fig. 11h). To confirm the function of our KLF gene regulatory network in human AUD (Fig. 5g), we constructed a GRN for these same TFs in our mouse model (Supplementary Fig. 11i). We found that combined KLF16, KLF 7 and KLF6 form a GRN that regulates 549 DEGs. We found 50 DEGs overlapped between the human and mouse GRNs including GWAS gene *CALN1* confirming its likely role in alcohol abuse. Taken together, these data support the functional significance of KLF TFs in regulating calcium and glutamatergic signaling pathways identified in our dataset.

## Cell-type specific variant fine mapping within the AUD gene regulatory landscape

In order to assess the role genetic risk variants play in AUD brain pathology, we performed linkage disequilibrium score regression (LDSC) for AUD and PAU (problematic alcohol use disorder includes individuals with AUD and alcohol related traits) across GWAS' including European ancestry, trans-ancestry and sex comparisons. We also included other commonly comorbid psychiatric disorders with AUD, and additional control traits to better understand the landscape of genetic risk within the CeA (Fig. 6a, Supplementary Data 33). We found significant enrichment for AUD and PAU GWAS signals in both INH and EXC neurons across all datasets. Only one GWAS (Audit-C, alcohol use disorders identification test-consumption)[14] showed significant enrichment in INH alone. Finally, we performed an LDSC enrichment by genic region and observed that AUD risk SNPs were enriched within introns of INH (Fig. 6b, Supplementary Data 34).

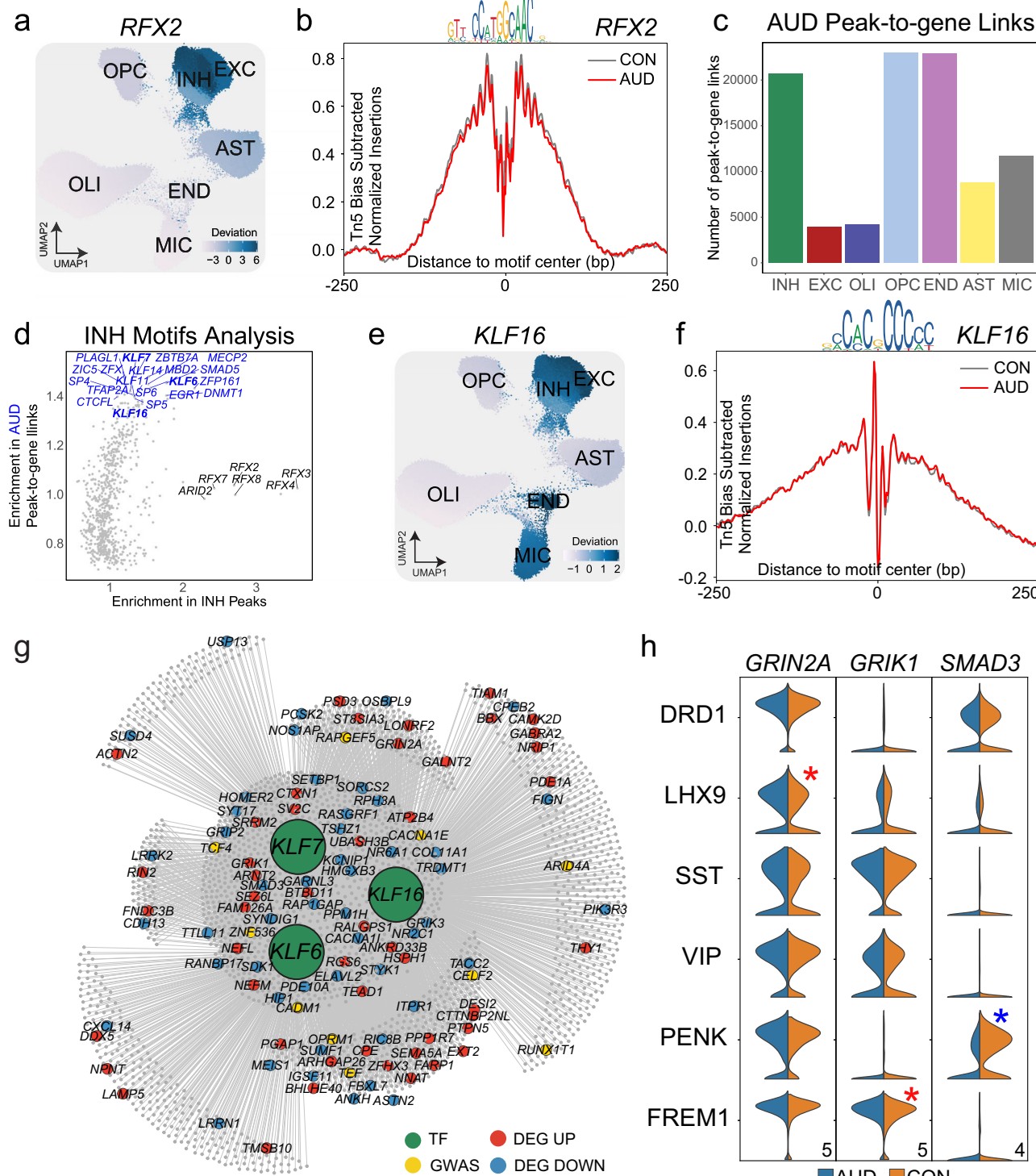

**Fig. 5 | Cell type-specific TF binding regulation in AUD. a** UMAP of *RFX2* chromVAR deviation scores show enrichment in INH neurons. **b** Tn5 bias-subtracted TF footprinting for *RFX2* in INH CON cells (gray) and INH AUD cells (red). TF motif logo is shown above the footprint. **c** Barplot of AUD peak-to-gene links. Number shows links that have a correlation difference greater than 0.5 between AUD and CON. Only peak-to-gene links with positive correlations were considered. **d** Scatterplot shows the motifs enriched in AUD-specific peaks (blue) and INH's marker peaks (black). *RFX2*, *RFX3*, and *RFX4* are examples of the marker motifs, while *KLF6*, *KLF7*, and *KLF16* are examples of the AUD-enriched motifs. **e** UMAP of *KLF16* chromVAR deviation scores show enrichment in INH neurons. **f** Tn5 bias-subtracted TF footprinting for *KLF16* in INH CON cells (gray) and INH AUD cells (red). The TF motif logo is shown above the footprint. **g** TF regulatory network showing the target genes for TFs *KLF6*, *KLF7*, and *KLF16* in INH. Peak-to-gene correlation > 0.9 was employed in generating the network. Top 10 GWAS and 50 Up & Down DEGs are shown. **h** Violin plot showing AUD versus CON expression of *GRIN2A* (FDR = 3.68e−6), *GRIK1* (FDR = 1.47e−6), *SMAD3* (FDR = 1.13e−5) in each inhibitory subtype with significant up (red asterisk) and down (blue asterisk) expression (one-sided Wald's test with corrections for multiple comparisons) (Numbers in the right bottom corner indicate maximum gene expression value for each gene).

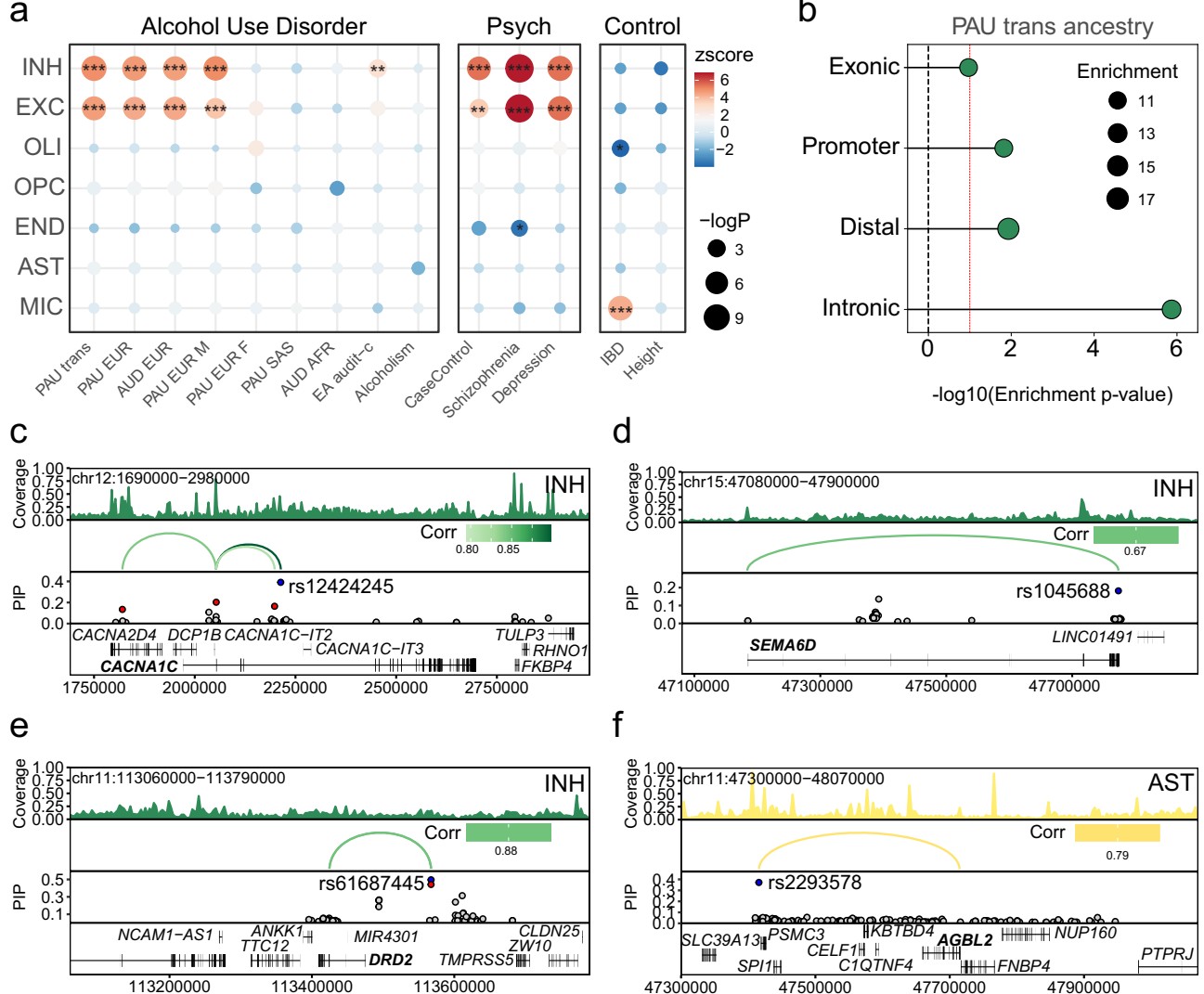

**Fig. 6 | Cell type-specific *cis*-regulation at AUD disease genetic risk loci. a** LDSC enrichment of various GWAS traits including Problematic Alcohol Use/Alcohol Use Disorder, psychiatric, and non psychiatric disorders in the snATAC-seq cell type peaks. FDR-corrected *P*-values are overlaid on the dotplot (two-sided z-test; *FDR < 0.05, **FDR < 0.005, ***FDR < 0.0005). **b** LDSC enrichment of the trait Problematic Alcohol Use trans-ancestry of the INH celltype peaks partitioned by genomic location (two-sided z-test). Red dashed line indicates a *P*-value of 0.10. **c-f** Cis-regulatory architecture at the following GWAS loci and cell types: *CACNA1C*, *SEMA6D*, *DRD2* in INH and *AGBL2* in AST. Union peak set was used to fine-map SNPs for PAU_EUR. For the fine-mapping track, we only plot SNPs with PIP > 0.01 for visualization purposes and only highlight peak-to-gene loops for SNPs with PIP > 0.09, indicated in red if they are within a peak, in blue if they are the lead within peaks, otherwise indicated in gray. The loops are colored based on the cell type that has the peak containing the lead peak variant. Chromosome coordinates are: *CACNA1C* (chr12:1690000–2980000); *SEMA6D* (chr15:47080000–47900000); *DRD2* (chr11:113060000–113790000); *AGBL2* (chr11:47300000–48070000). Corr = Correlation.

Because CREs regulate expression of nearby genes (functioning as promoters and enhancers), we reasoned that GWAS risk variants within them were more likely to be biologically impactful on the AUD brain transcriptome. We performed fine-mapping using SuSiE[53] to identify risk variants with epigenetic information from our single cell ATAC peaks for CREs across all cell types. We integrated cell type specific CRE information with 131 PAU genome-wide risk genes from the largest alcohol-related GWAS[54] across all cell types. For each gene we calculated posterior inclusion probability to both identify and/or confirm lead SNPs from the original study. We identified 223 credible lead SNPs (168 located within our open chromatin peakset) for 98 genes across cell types (Supplementary Data 35). We confirmed 52 SNPs previously reported (7 existing within our chromatin peakset) for 63 genes across cell types (Supplementary Data 36). Of the 131 GWAS DEGs, we found lead SNPs for 84 in INH. We identified credible lead SNPs for several genes implicated in neurotransmitter release and excitatory/inhibitory

(E/I) balance including *CACNA1C* (rs12424245 PIP = 0.39, *P*-value = 1.29e−34) (Fig. 6c), *SEMA6D* (rs1045688 PIP = 0.18, *P*-value = 2.41e−13) (Fig. 6d), *PPP1R13B* (rs11160760 PIP = 0.33, *P*-value = 3.79e−40) (Supplementary Fig. 12a), and *BCL11B* (rs1405238 PIP = 0.67, *P*-value = 3.47e−57) (Supplementary Fig. 12b). We also confirm the lead SNP for *DRD2* (rs61687445 PIP = 0.50, *P*-value = 6.58e−31) in INH neurons (Fig. 6e). In AST, of the 131 GWAS DEGs we found 74 lead SNPs. We highlight the gene *AGBL2* in AST (Fig. 6f) where we identified a credible lead SNP (rs2293578, PIP = 0.37, *P*-value = 2.23e−21). Finally, of the 131 GWAS DEGs we found lead SNPs for 84 in MIC. We highlight the risk gene *NF1* in MIC (Supplementary Fig. 12c) where we identified a credible lead SNP (rs2214538, PIP = 0.22, *P*-value = 1.12e−17). The rs2214538 SNP was the only one within a MIC ATAC peak and maintain significant predicted looping. By fine-mapping PAU risk variants within cell type-specific ATAC data with posterior probability, we have nominated or

confirmed putative causal risk loci and highlight these as candidates for additional functional genomic follow up.

## Discussion

In this work, we present the largest single cell transcriptomic and epigenomic case-control analysis of alcohol use disorder. We produced a high-resolution dataset of the central amygdala featuring 174,188 single cell transcriptomes and epigenomes from 50 donors and annotated seven neuronal and eight non-neuronal cell types allowing us to investigate cell type-specific dysregulation of gene expression in the context of alcohol addiction. We observed changes in gene expression across all cell types, however the majority of changes occurred in INH which are the most abundant neuronal subtype in the CeA. We were able to capture a high number of DEGs, which were largely cell type-specific. Current methods for identifying single cell DEGs are often plagued by numerous false positives[55]. To ensure that our findings are robust, we analyzed our data with both the Pegasus[40] based Wilcoxon Test and the more conservative MAST method (including covariates). We report here, DEGs that intersected between the two tests.

We identified significant enrichment of AUD and PAU risk genes in our DEGs. Of the 131 risk genes in the largest alcohol GWAS to date[16], 19 were significant DEGs. Of these, the majority (52.6%) were expressed in INH neurons and their subtypes. Following INHs, most DEGs were found in EXNs, AST and END cells. Astrocytes modulate motivational behaviors in response to drugs and ethanol[56]. Several studies in rodent models have found that alcohol increases calcium activity in AST in the central amygdala, which leads to increased EXC activity in neurons and more alcohol consumption[57]. Consistent with previous work, several nominally significant pathways in AUD ASTs regulate calcium signaling (Supplementary Data 12, 13). Chronic alcohol use can impair END cell function by reducing nitric oxide bioavailability, and increase oxidative stress, and inflammatory or oxidative injury to the endothelium[58], pathways significantly enriched in ENDs.

Astrocytes play a critical role in the biology of AUD, especially in the amygdala where their role in emotional and reward behaviors are well understood. Previous studies have found disruption in gene expression, alterations in glutamate uptake and inflammatory signaling in astrocytes in alcohol addiction and these changes likely regulate neuronal hyperexcitability and synaptic activity[59,60]. In the amygdala specifically, dysfunctional astrocyte activity has been linked to heightened anxiety, stress reactivity, and craving, all core features of AUD[61]. We found numerous DEGs in the astrocytes of the AUD CeA and enrichment of CREB signaling genes (Fig. 3d). FGFR3 has been shown to be upstream of CREB phosphorylation (activation)[62] and was significantly down-regulated in our AUD astrocytes (Supplementary Fig. 9g). Disruptions in CREB signaling likely result in alterations in neurotransmitter release and calcium dynamics in neighboring neurons consistent with the other molecular changes we observe in the CeA.

The AUD-related alterations in CeA interneurons, specifically PENK INH neurons, are particularly striking. The central amygdala is the primary synaptic output nucleus of the amygdala[63], projecting to the hypothalamus[64] and bed nucleus of the stria terminalis[63]. It is primarily made up of INH neurons and our transcriptomic atlas identified six INH subtypes. Despite being one of the smallest populations of INH, we found the most DEGs in PENK INH and extensive peak to gene ATAC links in this subtype. Proenkephalin is an endogenous opioid and this system has been implicated in both alcohol use and the development of AUD symptoms[65]. Both opioid agonists and antagonists can modulate alcohol consumption and are currently being developed for AUD pharmacotherapy[65,66]. Alcohol itself does not bind to *mu* opioid receptors but they are responsible for neurotransmitter release related to the pleasurable, rewarding effects of alcohol consumption[67]. Additionally, we found a significant increase in the metabotropic glutamate

receptor 8 (*GRM8*) in PENK INH neurons, suggesting increased glutamatergic signaling that we hypothesize may be disrupting E/I balance in the AUD CeA which could lead to increased alcohol consumption. E/I balance is critical for homeostatic regulation in the brain. Glutamatergic and inhibitory signaling imbalances in the central amygdala have been linked to alcohol related behaviors including chronic use and withdrawal[52,68]. We found additional lines of evidence pointing to disrupted E/I balance across all INH neuronal subtypes. We found significant increases in *CALN1* transcripts in LHX9 INH where there were also extensive peak-to-gene links in chromatin assembly pointing to epigenomic regulation of calcium-binding genes. An increase in calcium-mediated, activity-dependent mechanisms may result from increases in glutamatergic transmission within AUD CeA. The increase in *GABRA2* transcripts in two INH subtypes, suggests the AUD CeA may be attempting to homeostatically upregulate GABAergic transmission to rebalance E/I.

Including nuclei-matched chromatin assembly profiles with our single cell transcriptomics provides the opportunity to identify *trans*- and *cis*-acting elements regulating AUD gene expression. We identified ~500,000 putative CREs, consistent with previous human brain reports[49,69]. Unsurprisingly, we found most peak to gene links with DEGs occurred in INH and EXC neurons, many at AUD GWAS loci including at the genes *GABRA2* and *CALN1*. We followed up these analyses to perform fine mapping across all PAU risk loci identified in a recent large GWAS study[16]. By integrating open chromatin regions of the genome, we were able to identify 223 credible lead SNPs for these risk loci, many of which are involved in glutamatergic and calcium signaling. We focused our efforts primarily on INH neurons as these cell types were the most affected on the transcriptomic level, but were able to identify additional credible SNPs in other cell types, such as MIC and AST.

Additional analysis of AUD chromatin assemblies provides the opportunity to identify *trans*-acting regulation of gene expression through integration of TF footprinting. We found significant enrichment of many TFs in our AUD open chromatin regions. Noticeably, we found numerous Kruppel-like TFs enriched including *KLF 6, KLF7, KLF11, KLF15* and *KLF16*. The Kruppel-like factor family of TFs control expression of genes related to metabolism[70] and their dysfunction have been implicated in metabolic syndromes such as diabetes and liver disease[71]. Further, previous studies have identified alcohol abuse related changes in several KLFs[72,73]. Gene regulatory network analysis of three KLFs revealed moderate overlap in downstream genes. Interestingly, many of these genes were cell type specific DEGs and PAU GWAS genes, suggesting *trans*-acting effects on both disease and risk gene expression during AUD. We were also able to confirm the presence of a GRN of *KLF6, 7* and *16* in our animal model of alcohol use (Supplementary Fig. 11). However, this was a short-term administration of ethanol and thus limits some of the interpretation of this finding.

The resource presented here complements several other genomic studies of the AUD brain[74–78] and offers a reframing of gene expression changes and epigenomic regulation within specific cell populations. Identification of transcriptional regulators and DEGs linking calcium activity and glutamatergic signaling with E/I imbalances in the central amygdala contextualize the possible outcomes of known genetic risk factors for AUD. In conclusion, this work provides additional avenues for future research to identify the mechanisms underlying this complex disorder.

## Methods

### Human tissue donors

Human postmortem brain tissue samples were obtained from the National Center for PTSD Brain Bank (with consent of next of kin) and the University of Pittsburgh Tissue Donation Program. Brain specimens were obtained during autopsies conducted at the Allegheny

County Medical Examiner's Office (Pittsburgh, PA) after consent for donation was obtained from the next-of-kin. Individuals were a mix of European– and African–American descent. Males and females were group-matched for age and PMI. Sociodemographic and clinical details are listed in Supplementary Data 1 and include the manner of death, alcohol-related phenotypes, tobacco use, additional psychiatric disorders, and illicit drug use. Inclusion criteria for AUD and CON cases were as follows: PMI < 48 h, age range >18 to <85 years. A total of 50 individuals (22 AUD: 9 females, 13 males; and 28 control donors without AUD: 17 females, 11 males) were used in this study. To control for the high comorbidity of mood disorders in our AUD sample (100% are comorbid for either major depression or PTSD) we included additional donors ($N = 15$, 53.5%) with no history of alcohol abuse or dependence but did have MDD or PTSD in our control group (control donors without AUD; Supplementary Data 1) to aid in identifying AUD-specific changes. The brain tissue was fresh-frozen and samples from the central nucleus of the amygdala[79] were collected (-5 mg) from each postmortem sample.

Psychiatric history and demographic information were obtained by psychological autopsies performed postmortem as well as a review of medical records and toxicology reports. Trained clinicians conducted structured interviews with informants (usually the next of kin) with knowledge of the deceased individuals. To avoid systematic biases, AUD and control cases from each source were characterized by the same psychological methods. Consensus DSM-IV diagnoses for each subject were made by trained clinicians who did not conduct the psychological autopsies. Severity ratings are also based on DSM-IV.

All brain tissue underwent physical examination by a board-certified neuropathologist at both the macroscopic and microscopic level. All tissue was screened for confounding neuropathologies, including amyloid plaques, Lewy bodies (Parkinson's disease and dementia with Lewy bodies), and transactive response DNA binding protein-43 (TDP-43) (ALS, frontotemporal dementia, and limbic predominant age-related TDP-43 encephalopathy). All cases were also evaluated at a gross (macroscopic) level to detect evidence of atrophy, trauma, or infarction. Importantly, cases were excluded if massive trauma occurred (from head injury) that severely damaged the tissue, if the tissue was damaged from a stroke involving a large portion of the brain, if the decedent was on a respirator for an extended period of time, if the donor had brain cancer, or if the donor had a history of HIV, AIDS, COVID-19, or other communicable disease.

## Brain nuclei isolation for multiome assay

Regions of interest were dissected on cryotome (leaflets of -100–300 micrometers) from frozen Central Amygdala and stored at −80 °C. Cell nuclei isolation from brain sections were treated similarly to already established protocols[80–82] with some modifications needed to have nuclei suitable for two separate assays: gene expression and ATAC. To avoid experimental bias, nuclei isolation was done by the same person blinded for the metadata for the particular sample. All reagents were molecular biology grade and sourced from Sigma unless stated otherwise. Small amounts of tissue (10–20 mg) were added into 1 ml of ice-cold lysis buffer ("Buffer A" is 250 mM sucrose, 25 mM KCl, 5 mM MgCl2, 10 mM Tricine-KOH (pH 7.8), protease inhibitors w/o EDTA (Roche), RNAse inhibitor (80 U/ml) (Roche), 1 mM DTT, 1% BSA (m/v) (Gemini Bio-Products, Woodland, CA), 0.3% NP-40 (v/v), 0.15 mM spermine, 0.5 mM spermidine, water–DTT, RNAse Protector, protease inhibitors, spermine, spermidine and NP-40 were added immediately before use. The suspension was transferred to 2 ml Dounce tissue homogenizer (Kimble) and lysed with constant pressure and without introduction of air with pestle A (2 × 5) and pestle B (4 × 5) in cycles of 5. The homogenate was strained through pre-wetted 40 μm tube top cell strainer (Corning). All subsequent centrifugation was performed in a refrigerated, bench-top centrifuge with swing-out rotor (Eppendorf, Hamburg, Germany). Lysates were centrifuged at 1000 g, 10 min, 4 °C, pellets were saved, and resuspended in 0.4 ml lysis buffer "Buffer A". Final 0.4 ml of solution was mixed with 0.4 ml (1:1) of Optiprep solution (Buffer "B" is iodixanol 50% (v/v), 25 mM KCl, 5 mM MgCl2, 20 mM Tricine-KOH (pH 7.8), protease inhibitors w/o EDTA, RNAse inhibitor (80U/ml), 1 mM DTT, water). The suspension (25% iodixanol final) was mixed 10x head over tail. The 2 ml tube was filled with 0.6 ml of 40% iodixanol cushion (appropriate mix of Buffer "A" and "Buffer "B"), then overlayed with 0.6 ml of 30% iodixanol (appropriate mix of Buffer "A" and Buffer "B"), and sample suspension was overlayed over at the top. The tubes were then centrifuged at 3000 g, for 30 min at 4 °C without brakes. After centrifugation ended the interphase 30%/40% ring was collected. Regions of interest were dissected on cryotome (leaflets of -100–300 micrometers) from frozen Central Amygdala and stored at −80 °C. Cell nuclei isolation from brain sections were treated similarly to already established protocols[80–82] with some modifications needed to have nuclei suitable for 10x Genomics Chromium Next GEM Single Cell Multiome ATAC + Gene Expression. To avoid experimental bias, nuclei isolation was done by the same person blinded for the metadata for the particular sample. All reagents were molecular biology grade and sourced from Sigma unless stated otherwise. Small amounts of tissue (10–20 mg) were added into 1 ml of ice-cold lysis buffer ("Buffer A" is 250 mM sucrose, 25 mM KCl, 5 mM MgCl2, 10 mM Tricine-KOH (pH 7.8), protease inhibitors w/o EDTA (Roche), RNAse inhibitor (80 U/ml) (Roche), 1 mM DTT, 1% BSA (m/v) (Gemini Bio-Products, Woodland, CA), 0.3% NP-40 (v/v), 0.15 mM spermine, 0.5 mM spermidine, water–DTT, RNAse Protector, protease inhibitors, spermine, spermidine and NP-40 were added immediately before use. The suspension was transferred to 2 ml Dounce tissue homogenizer (Kimble) and lysed with constant pressure and without introduction of air with pestle A (2 × 5) and pestle B (4 × 5) in cycles of 5. The homogenate was strained through pre-wetted 40 μm tube top cell strainer (Corning). All subsequent centrifugation was performed in a refrigerated, bench-top centrifuge with swing-out rotor (Eppendorf, Hamburg, Germany). Lysates were centrifuged at 1000 g, 10 min, 4 °C, pellets were saved, and resuspended in 0.4 ml lysis buffer "Buffer A". Final 0.4 ml of solution was mixed with 0.4 ml (1:1) of Optiprep solution (Buffer "B" is iodixanol 50% (v/v), 25 mM KCl, 5 mM MgCl2, 20 mM Tricine-KOH (pH 7.8), protease inhibitors w/o EDTA, RNAse inhibitor (80U/ml), 1 mM DTT, water). The suspension (25% iodixanol final) was mixed 10x head over tail. The 2 ml tube was filled with 0.6 ml of 40% iodixanol cushion (appropriate mix of Buffer "A" and "Buffer "B"), then overlayed with 0.6 ml of 30% iodixanol (appropriate mix of Buffer "A" and Buffer "B"), and sample suspension was overlayed over at the top. The tubes were then centrifuged at 3000 g, for 30 min at 4 °C without brakes. After centrifugation ended the interphase 30%/40% ring was collected (300 microliters). 500 microliters of "ATAC Lysis Buffer" was added to the sample, mixed well and left on ice for 5 min. "ATAC Lysis Buffer" is: 10 mM Tris-HCl (pH 7.4), 10 mM NaCl, 3 mM MgCl2, 0.01 % (v/v) Tween-20, 0.01% (v/v) NP-40, 0.001% (v/v) Digitonin, 1 mM DTT, RNAse inhibitor (80 U/ml) (Roche), 1% (m/v) BSA, Nuclease-free Water. After 5 min, 500 microliter "ATAC Wash Buffer" was added to ATAC sample, mixed well and left on ice for 5 min. "ATAC Wash Buffer" is: 10 mM Tris-HCl (pH 7.4), 10 mM NaCl, 3 mM MgCl2, 0.1 % (v/v) Tween-20, 1 mM DTT, RNAse inhibitor (80 U/ml) (Roche), 1% (m/v) BSA, Nuclease-free Water. The samples were centrifuged at 1000 g, 10 min, 4 °C, supernatants were carefully and completely removed. After last centrifugation the samples were resuspended in "Diluted Nuclei Buffer" (1X Nuclei Buffer (PN-2000207), 1 mM DTT, 1 U/microliter RNase Protector, water) counted on hemocytometer at the concentration of 3.3 million/mL and submitted to YCGA where the protocol was followed for targeting 10000 nuclei. Libraries were

sequenced with paired-end 150 bp reads on an Illumina NovaSeq 6000 to a target depth of up to 500 million read pairs per sample for gene expression (RNA), and up to 500 million reads per sample for ATAC.

## Animals

All animal care and procedures were performed according to the ethical guidelines of the Institutional Animal Care and Use Committee at Yale University School of Medicine and Yale Animal Resources Center. Adult 2-month-old male and female C57BL/6 mice ($n = 8$ mice in total) were used for this study. Until treatment, all mice were group housed and maintained in standard environmental conditions (23 °C; 12 h–12 h light–dark cycle) with ad libitum food and water.

## Ethanol treatment

Mice were delivered at seven weeks of age and allowed two weeks to acclimate to the vivarium, where they were housed in a reverse light-dark cycle housing room. Two hours into the dark cycle, body weights were measured. Three hours into the dark cycle, mice received two sequential intraperitoneal injections each with a 10 ml/kg injection volume: the first was a solution of 5% DMSO in saline directly followed by 2 g/kg ethanol dissolved in water ($n = 2$ M, 2 F) or saline ($n = 2$ M, 2 F). Animals were randomly assigned to control or ethanol treated groups. Three hours following these injections, mice were euthanized by rapid decapitation and brains removed and immediately snap frozen on dry ice and then stored at −80 °C. For tissue analyses, frozen brains were sliced into 2-mm coronal sections on a metal brain matrix (Plastics One, Roanoke VA) and amygdala samples were collected bilaterally with a 1-mm tissue punch (Fine Science Tools, Foster City, CA). Bilateral punches were collected from a coronal section targeted at −1.43 mm from bregma, according to[83]. Nuclei were isolated from rodent amygdala tissue as described in Brain Nuclei Isolation for Multiome methods section.

## Fluorescence Immunohistochemistry

Immediately following autopsy, tissue blocks for immunohistochemistry were cut and submerged in 4% paraformaldehyde for 48 hrs at 4 °C, followed by cryoprotection in sucrose. Sections were cut on a microtome at 40 um and stored in cryoprotectant at −20 °C until use.

Following washes in phosphate buffer, tissue sections were pretreated for antigen retrieval (0.01 M sodium citrate for 45 min at 90 °C), followed sequentially by incubation in 1% sodium borohydride (30 min), PB wash, incubation in 0.3% Triton X-100 (30 min), blocked in 20% normal donkey serum (1.5 h), and incubated at 4 °C for 72 hr with the following primary antibodies: goat anti-Gad1 1:100 dilution (R&D Systems AF2086), rabbit anti-PENK 1:500 dilution (Atlas Antibodies HPA013138), anti-Rb 647; rabbit anti- FREM1 1:50 dilution (Proteintech 13086-1-AP). Secondary antibody incubations were overnight at 4 °C with anti-goat Alexa fluor-488 or anti-Rb Alexa Fluor- 647 (Invitrogen, Life Technologies, Carlsbad, CA). Following washing, sections were incubated in DAPI and mounted with ProLong Glass Antifade reagent (Life Technologies, Carlsbad, CA). Z-stack images were collected through the depth of each section on an Olympus FV-3000 confocal scanning laser microscope using a 20×, 0.75 na objective.

## Acetylcholinesterase staining

A modification of the method of Lim et al., 2004 was used for acetylcholinesterase staining to aid in delineating amygdala subregions[84]. Briefly, slide mounted sections adjacent to sections used for marker identification were incubated for 2 h in 0.12% acetylthiocholine iodide, 0.075% glycine, 0.05% cupric sulfate and 0.68% sodium acetate. Slides were washed 3 times and incubated in 0.77% sodium sulfide nonahydrate for 15 min. Following 3 washes, slides were exposed to 1% silver nitrate for 10 min and washed again. Slides were dehydrated in ethanol, cleared and mounted.

## RNAScope in situ hybridization

Blocks of fresh frozen human amygdala from three AUD subjects and three controls were sectioned at 10 µm and stored at −80 °C. RNA-Scope in situ hybridization was performed with the RNAscope Multiplex Fluorescent Detection Kit v2 (Cat# 323100, ACD, Newark, CA, USA) according to manufacturer's instructions. For each subject, one brain section was stained. Briefly, tissue sections were fixed with 10% neutral buffered formalin (Cat # 15740, Electron Microscopy Sciences, Hatfield, PA, USA) for 30 min at RT, series dehydrated in ethanol, pretreated with hydrogen peroxide for 10 min, and treated with protease IV for 30 min. Sections were incubated with probes for *PENK*, *GRM8*, *NCAM*, *CALN1*, and *TCF4* (Cat# 548301-C2, 563351, 421461, 1317671-C1, and 557411 ACD, Newark, CA, USA). Probes were fluorescently labeled with TSA dyes (TSA-Vivid 650, Cat#7536, and TSA-Vivid 570, Cat#7535, Tocris Biosciences, Bristol, UK). Nucleus were labeled with DAPI (4′,6-diamidino-2-phenylindole).

Fluorescent images were acquired in z-series using an Olympus FV3000 confocal microscope with 60× 1.42NA objectives. For each brain section, images were taken from 3 regions of interests within the central amygdala. All images for the same combination of probes were acquired with the same imaging settings and laser power intensities. Subsequent automated quantification was performed utilizing the *dotdotdot* algorithm for MATLAB[85]. DAPI stained nuclei, transcripts and lipofuscin were segmented from the background. The segmented RNA transcripts were masked with segmented lipofuscin before the colocalization of transcripts and nuclei were counted. Statistics were performed with Prism 10 (Graphpad).

## Single-nucleus multiome processing

**Count matrix generation.** We ran CellRanger ARC (v2.0.2) with the hg38 genome as the reference to process the initial reads from the raw FASTQ files. ***Quality control and filtering***. We used Signac[86] (v1.11.0), a comprehensive R package for the analysis of single-cell chromatin and gene expression data, to preprocess each snMultiome sample. Signac creates a Seurat object containing both RNA and ATAC assays. For quality control, we filtered cells using the following parameters: nCount_ATAC > 1000, nCount_RNA > 200, and TSS.enrichment > 2.

**RNA – Doublet detection.** Next, doublets were identified using a combination of two computational methods: Scrublet[87] and DoubletDetection[88] and removed from each sample. ***Sample aggregation***. We merged high-quality cells and high-quality samples in Pegasus (v1.5.0)[40], a python tool for analyzing transcriptomes of single cells, resulting in a total of 55 samples (Supplementary Fig. 13). After aggregating these samples into a single data object, we filtered cells based on the following criteria: at most 10% mitochondrial genes, at least 200 genes, and at least 500 UMIs. Mitochondrial, sex, and ribosomal genes were excluded, and only the robust genes were included in the final gene set. Five samples were further removed due to a surprisingly high EXC neuron composition, not expected in the central amygdala. The final snRNA-seq data object had 183,391 nuclei and 26,919 genes. ***Clustering and annotation***. After cell and gene filtration, the data was transformed into logarithmic space by normalizing each cell by total counts. Next, the top 2000 highly variable features were found and principal component analysis (PCA) was applied on the highly variable features using the top 50 components. Batch correction was performed with Harmony. The top 50 components were then used to build a k-nearest-neighbors graph with k = 100 neighbors, and Leiden clustering was used to identify cell clusters. After the first round of clustering, marker genes were used to inspect and remove clusters that had mixed or unmatched marker gene expression, and another round of clustering was performed. Cell type annotation was adapted

using markers from Ma et al. with 7 cell types[80]. Further subtypes were discovered through either NSForest de Novo or confirmed via existing literature on the amygdala[36–38]. Most notably, we discovered 6 INH subtypes: *DRD1*, *LHX9*, *SST*, *VIP*, *PENK*, and *FREM1*.

**ATAC—Quality control and filtering.** Continuing from the 183,391 nuclei in the RNA QC step, we created Arrow files for each sample with cells filtered based on the quality control parameters filterTSS=4 and filterFrags=1000 in ArchR (v1.0.3)[89], an R package for analyzing single-cell ATAC-seq data with multi-omic support. ArchR considers three characteristics: 1) the number of unique nuclear fragments, 2) the signal-to-background ratio, and 3) the fragment size distribution. Due to nucleosomal periodicity, we expect to see depletion of fragments that are the length of DNA wrapped around a nucleosome, around 147 bp. We used the addDoubletScores function to infer potential doublets that can confound downstream results. ***Clustering and annotation.*** The addIterativeLSI function was used to implement iterative LSI dimensionality reduction. Cell type annotations were transferred from the labels in the RNA modality, leveraging the multi-omic nature of our dataset. We visualized the embedding using addUMAP and plotEmbedding. We produced a final embedding of 174,188 high-quality nuclei in both RNA and ATAC modalities (Supplementary Fig. 13, Supplementary Data 37).

### Differential gene expression analysis
snRNA-based differential expression analysis was performed for each cell type and the 6 INH subtypes using four tests: 1) parametric-based MAST[39] with covariate correction including covariates such as age, sex, ancestry, PMI, and RIN, 2) permutation-based Wilcoxon Rank Sum Test using the de_analysis function from Pegasus, 3) parametric-based MAST[39] with covariate correction including covariates such as age, sex, ancestry, PMI, RIN, and additional alcohol clinical covariates 4) pseudobulk using DESeq2[41] (v1.46.0) with covariate correction. Only genes expressed in at least 5% of the cells, the robustly expressed genes, were included for these tests. The FDR significantly differentially expressed genes (DEGs) were those that passed the thresholds of fold change (FC) > 1.2 and FDR < 0.01 and overlapped in the same direction for MAST and Wilcox. For INH, we used the FC value of the subtype that had the greatest magnitude for analyses. These DEG lists were utilized in downstream analyses such as TF-CRE-DEG links.

### Peak calling on snATAC clusters
Pseudo-bulk replicates were created for each cell type with the addGroupCoverages function to resolve inherent sparsity in the snATAC-seq data. MACS2[48] was then used to call peaks on each pseudo-bulk replicate. ArchR uses an iterative overlap peak merging procedure with the addReproduciblePeakSet function to create a final peak set of fixed-width peaks using the following procedure: 1) peaks are first ranked by their significance, 2) the most significant peak is retained and any peak that directly overlaps with the most significant peak is removed from further analysis, and 3) of the remaining peaks, this process is repeated until no more peaks exist. ArchR analyzes all the pseudo-bulk replicates from a single cell type together, performing the first iteration of iterative overlap removal. It then checks to see the reproducibility of each peak across pseudo-bulk replicates and only keeps peaks that pass a threshold indicated by the reproducibility parameter. This resulted in a single merged peak set for each of the seven cell types. This procedure is then repeated to merge the cell type-specific peak sets by renormalizing the peak significance across the different cell types and performing the iterative overlap removal, resulting in a single merged peak set of fixed-width (501 bp) peaks, denoted the union peaks. We utilized both cell type-specific peaks and the union peaks depending on the analysis we performed. The addPeakMatrix function adds the Peak Matrix using the union peaks to the data object, which is used for

downstream analyses such as finding peak-to-gene links and performing TF motif enrichment analysis.

### Characterization of CREs
Since snMultiome already simultaneously profiles the gene expression and chromatin accessibility of the same nuclei, no integration between the two modalities was needed. Directly using the GeneExpressionMatrix, we identified peak-to-gene links using the addPeak2GeneLinks function, which leverages integrated snRNA-seq data to look for correlations between peak accessibility and gene expression. We set maxDist=1,000,000 to consider long-range interactions. Then we identified putative enhancers or *cis*-regulatory elements (CREs) based on the thresholds: peak-to-gene correlation > 0.4 and FDR < 0.05.

### TF motif enrichment analysis and TF regulatory network construction
TF motif enrichments were calculated to predict which regulatory factors are most active in a given cell type. The addMotifAnnotations function indicates motif presence in the union peaks using a binary matrix, and the peakAnnoEnrichment function tests either cell type-specific marker peaks or differential peaks for enrichment of various motifs. Besides the previous TF motifs enrichments, the R package chromVAR predicts enrichment of TF activity on a per-cell basis while controlling for known technical biases. The addDeviationsMatrix function, which adds the MotifMatrix to the data object, was used to compute deviation *z*-scores for each motif by comparing the number of fragments that map to peaks containing the motif to the expected number of fragments in a background peak set that accounts for confounding factors such as GC content bias, PCR amplification, and Tn5 tagmentation. Then we used the getPositions, addGroupCoverages, and getFootprints functions to perform TF footprinting analysis with pseudo-bulk aggregates of cells in the same cell type. In the plotFootprints function, ArchR performs normalization by subtracting the Tn5 bias from the footprinting signal. We plot the footprinting results for TFs *RFX2* and *KLF16* comparing AUD and CON in INH neurons. We then created TF regulatory networks based on the TF-CRE-DEG links. We identified several TFs with high enrichment and plotted their target genes using the R package igraph (v1.2.6), where each vertex represents a TF or target gene, and each edge represents a linked enhancer binding event, overlaying additional information onto the network such as DEG (up/down) or AUD GWAS gene (True/False) status. For the TF networks, we used more conservative correlation cutoffs of 0.9 for visualization purposes.

### Disease-specific peak-to-gene calculation
To combat the sparsity of peak data, which makes it difficult to calculate disease-specific differences, we focused on peak-to-gene linkages for our multi-omic data. Cell-type-specific peak-to-gene links were first calculated for each of the 7 canonical cell types via the cellsToUse parameter in the addPeak2GeneLinks function. Then, for each cell type, peaks with strong peak-to-gene correlation in AUD (corr_AUD > 0.5) and also a large difference between AUD and Control (corr_diff > 0.5) were kept. Finally, motif enrichment was performed on these subsetted peaks for each cell type to highlight disease-specific motifs.

### Estimating GWAS enrichment using cell type-specific peaks
To estimate heritability of a variety of complex traits, we used LDSC (v1.0.1)[90]. GWAS summary statistics for Problematic Alcohol Use/Alcohol Use Disorder, psychiatric, and non psychiatric disorders were correctly formatted with munge_sumstats.py and lifted to hg38 coordinates using UCSC liftover. Cell type-specific peaks, peaks from the disease-specific peak-to-gene calculation, and INH peaks separated by

genomic region were formatted for LDSC using make_annotation.py, and LD scores were computed for each set using ldsc.py. Benjamini-Hochberg multiple-testing correction was applied to the enrichment *P*-values.

### GWAS fine-mapping

**Peak enrichment scores as prior weights.** We performed genetic fine-mapping with the sum of single-effects (SuSiE) regression model[53] to infer risk variants with epigenetic information from our single-nuclei ATAC peaks. With some prior information on SNPs, we can improve the accuracy of detecting the causal variants[91]. For SNPs inside ATAC peaks, we used the peak enrichment scores (−log10pval) which indicate the strength of each peak, as the prior weights for SuSiE to prioritize SNPs with strong ATAC signals. For SNPs outside ATAC peaks, we set the prior weights to 0.1. We then subset the union peaks to the CREs (peak-to-gene correlation > 0.4 and FDR < 0.05) to perform fine-mapping in SuSiE. We performed SuSiE regression with the susie_rss function using AUD GWAS summary statistics (PAU EUR)[16]. SuSiE calculates the posterior inclusion probability (PIP), the probability for a given SNP being causally associated with the trait of interest, for each variant. The LD information was derived from the 503 European samples of the 1000 Genomes Project. ***Input files.*** We used the 1000 Genomes Project (1KG) Phase 3 dataset[92] as the reference panel for our analysis. This dataset can be accessed via the Resources section on the PLINK 2.0 website at Resources-PLINK2. 0(cog-genomics.org). We selected the European samples based on the super-population information provided by 1KG and excluded all duplicate and ambiguous SNPs. We applied quality control to the 1KG data with European ancestry using PLINK[93]. To be more specific, we only included SNPs with a 99% genotyping rate and the minor allele frequency (MAF) no less than 0.005. We also excluded samples with more than 5% missing genotypes and markers that failed to pass the Hardy-Weinberg test (PLINK command: --geno 0.01 −hwe 1e-10 −mind 0.05 −maf 0.005). The final reference panel includes 503 non-overlapping European samples, genotyped at 8,190,311 SNPs. To obtain the coordinates for risk SNPs, we turned to the UCSC Genome Browser[94,95] and focused on the human genome version GRCh38/hg38.

### Reporting summary

Further information on research design is available in the Nature Portfolio Reporting Summary linked to this article.

## Data availability

The snMultiome data generated in this study have been deposited in the Zenodo database [https://doi.org/10.5281/zenodo.17656668]. Datasets are available from the corresponding author and requests may also be submitted to https://www.research.va.gov/programs/tissue_banking/ptsd/ and referencing this paper. The processed data generated in this study are provided in the Supplementary Information/Source Data file. Source data are provided with this paper.

## Code availability

All code used in this study is freely available online and can be found at https://github.com/mjgirgenti/AUDsnCEA.

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

## Acknowledgements

We would like to express our gratitude to the National Center for PTSD Brain Bank, the University of Pittsburgh Brain Tissue Donation Program, and the NIH NeuroBioBank whose efforts led to the donation of the postmortem tissue used in these studies. We are also indebted to the generosity of the families of the decedents, who donated the brain tissue used in these studies. The research reported here was supported by the Department of Veterans Affairs, Veteran Health Administration, VISN1 Career Development Award, a Brain and Behavior Research Foundation Young Investigator Award, an American Foundation for Suicide Prevention Young Investigator Award, NIH grants R01AA031017 and DP1DA060811 to M.J.G., R01HG012572, R01DA063316 to J.Z. and P50AA012870 J.H.K. We thank the Keck Microarray Shared Resource (KMSR) and Yale Center for Genome Analysis (YCGA) at Yale university for their assistance with snMultiome sequencing. This work was supported with resources and use of facilities at the VA Connecticut Health Care System, West Haven, CT, the Durham VA Healthcare System, Durham NC, and the VA Boston Healthcare System, Boston, MA, USA and the National Center for PTSD, U.S. Department of Veterans Affairs. This work was funded in part by the State of Connecticut, Department of Mental Health and Addiction Services. The views expressed here are those of the authors and do not necessarily reflect the position or policy of the US Department of Veterans Affairs (VA) or the U.S. government or the views of the Department of Mental Health and Addiction Services or the State of Connecticut.

## Author contributions

C.L., J.Z., and M.J.G. conceived the project and designed the experiments. C.L. and M.J.G. wrote the manuscript. C.D., M.S., R.T., D.M., J.L., G.T., J.C., and A-N.S., generated all of the data. C.L., A.H., X.Z., S.X., J.W., T.N., Y.Liu., H.L., Y.D., Z.D., Y.Lei., Y. Lin, K.X., H.Z., J.Z., and M.J.G. oversaw all bioinformatics analyses. H.Zhou and J.G. contributed GWAS summary data and analysis. J.R.G., D.A.L, P.E.H., J.H.K., H.Zhou, S.T., J.T. and A.C. contributed to study design. All authors contributed to the preparation of the manuscript.

## Competing interests

J.H.K. has consulting agreements (less than US$10,000 per year) with the following: Aptinyx, Inc. Biogen, Idec, MA, Bionomics, Limited (Australia), Boehringer Ingelheim International, Epiodyne, Inc., EpiVario, Inc., Janssen Research & Development, Jazz Pharmaceuticals, Inc., Otsuka America Pharmaceutical, Inc., Spring Care, Inc., Sunovion Pharmaceuticals, Inc.; is the co-founder for Freedom Biosciences, Inc.; serves on the scientific advisory boards of Biohaven Pharmaceuticals, BioXcel Therapeutics, Inc. (Clinical Advisory Board), Cerevel Therapeutics, LLC, Delix Therapeutics, Inc., Eisai, Inc., EpiVario, Inc., Jazz Pharmaceuticals, Inc., Neumora Therapeutics, Inc., Neurocrine Biosciences, Inc., Novartis Pharmaceuticals Corporation, PsychoGenics, Inc., Takeda Pharmaceuticals, Tempero Bio, Inc., Terran Biosciences, Inc..; has stock options with Biohaven Pharmaceuticals Medical Sciences, Cartego Therapeutics, Damona Pharmaceuticals, Delix Therapeutics, EpiVario, Inc., Neumora Therapeutics, Inc., Rest Therapeutics, Tempero Bio, Inc., Terran Biosciences, Inc., Tetricus, Inc.; and is editor of Biological Psychiatry with income greater than $10,000. The remaining authors declare no competing interests.
