## [Transparent Peer Review file · Nature Communications]

Central amygdala single-nucleus atlas reveals chromatin and gene transcription dynamics in human alcohol use disorder

Corresponding Author: Dr Matthew Girgenti

Version 0:

Reviewer comments:

Reviewer #1

(Remarks to the Author)

The manuscript by Lee et al., reports an interesting and important multi-omic single nucleus profiling of postmortem human amygdala from donors with alcohol use disorder (AUD). The study aims to identify cell-type specific dysregulated gene regulatory programs. The approach is novel, and the study could improve our understanding of the molecular underpinnings of AUD; however, the design, the analysis and the reporting contain significant shortcomings that should be addressed.

Information on the metadata of the donors are significantly lacking, the authors do not report on the cause of death, lifetime consumption of alcohol, co-use of other substances of abuse (other than tobacco), presence or absence of psychiatric disorders, and related medications. All of which should have been included as covariates in the differential expression analyses.

A major limitation of the study is the lack of validation either in bulk tissues or alternative methods confirming cell specificity. Single cell transcriptome approaches are strengthened by including some experimental validations since many of the findings are computationally based.

There is a discrepancy between the reported method and the nuclei isolation protocol, 10x Genomics Chromium Next GEM Single Cell Multiome ATAC + Gene Expression profiling which simultaneously profiles ATAC and RNA sequencing in the same nucleus and the reported nuclei isolation method implies producing two aliquots from each sample one of which is processed for single nucleus ATAC and one for gene expression profiling.

The profiling yielded 183,391 cells from 50 samples (average 3600 nuclei per sample), the process as described targets 10,000 cells. Details should be provided whether the nuclei were eliminated during quality control, and report on quality control metrics for individual samples, or cells were lost during the library preparation process. How many cells were recovered from each sample?

How many cells were taken out with Scrublet and DoubletDetection? ~183,000 cells is quite low for 55 samples. What does the barcode rank plot look like across all samples, authors denote high and low quality but don't show QC metrics for these samples.

There is a discrepancy in reporting of the Wilcoxon test (`de_analysis` function from Pegasus in the methods and Seurat in the discussion section). The scripts used for performing MAST was not available at the time of review since link for the Github is not working. Please confirm active links to provide access to the scripts used in the study.

MAST and wilcoxon rank sum testing are quite lenient differential methods in single cell studies. This is evident in the higher fold change and FDR cutoffs used in the study. Were these directly compared to pseudo bulk methods? Sample size can have unexpected impacts on most current DEG methods. Using two methods and then overlapping does help ensure that what DEGs are being identified are consistent but it would be good to see pseudo bulking as well to determine the overlap between single nuclei based methods and a separate, more conservative approach for DE.

P values should be reported for proportional change showing no difference in cell types between AUD and Control.

It is stated that microglia and macrophages were found but do not show markers to differentiate and collapse them further in the paper. F13a1 is not exclusive to macrophages and can be representative of monocyte-derived infiltrating macrophages and in some cases microglia. Cd44, CD169 and Siglech have been reported as good marker for infiltrating macrophages rather than resident microglia, in addition to higher expression of CD9. What is the expression of a more robust microglia marker like TMEM119 or P2RY12? The identification of a true macrophage population is not possible without extensive markers and reporting the differences in key microglia homeostatic genes.

The authors report sex specific AUD changes and provide supplementary tables for male AUDvsCON and female AUDvsCON; however, there was no discussion of how the analysis were performed and specifics were not provided. Given the sample size this analysis is likely underpowered but potential sex differences should be examined and noted in the text.

Authors discuss the changes in chromatin accessibility that were linked into differentially expressed genes, these are potentially important findings, and additional details should be provided discussing instances where the changes of chromatin accessibility did not match observed gene expression changes and vice versa. Additional discussion of these findings is warranted.

Additional detail and discussion about the changes in cell types other than inhibitory neurons should be provided and any overlaps between the identified differentially regulated elements in these different cell types. This could be presented in the form of a potential supplemental table.

The authors note a nominally significant up regulation of OPRM1 in INH neurons. This finding is somewhat surprising since OPRM1 is often not dysregulated in transcriptome studies specifically examining OUD. The donor metadata should be examined for comorbid opioid use or reported OUD/MMD medication usage that could potentially explain this interesting finding.

Minor comments

Line 96: (mean reads/nuclei), how it is written it suggests that this was the target sequencing depth.

Figure 4b is not referenced in the text

(Remarks on code availability)

Links to GitHub scripts were not working for this reviewer.

Reviewer #2

(Remarks to the Author)

The manuscript by Lee et al provides insight into alcohol use disorder in humans by investigating single cell chromatin and gene expression data from post-mortem brain donors. The paper is well written and provides new data to the field. I'm not fully comfortable to provide an in-depth review of the bioinformatic analyses and therefore focus on clinical aspects primarily.

The title is slightly incorrect because the technique used is single *nucleus* multiome.

I'd suggest including the sample size in the abstract.

Line 89: A quick google search identifies snRNAseq studies in human CeA, I may be wrong.

The clinical spectrum includes individuals with alcohol abuse and alcohol dependence, two very different phenotypes. What is the rationale and justification to combine both beyond the notion that DSM-V collapsed those two (can they be combined in this dataset given that the clinical heterogeneity within AUD is large)? The donor's age at death is young, what led to their demise? Do you have information on the severity of AUD (mild, moderate, severe) and the timeline across life? Did donors consume alcohol at death (neurotox data)? Do you have information on the neuropathological assessments of the donors (cases and controls), especially for those >60yrs [neurodegen panel]?

Looking at the Extended Data Fig 1a, the details given regarding the post-mortem samples is very limited. The U Pitt Brain Bank has a large amount of data on their donors that is critical to include and to evaluate the data and this manuscript. For example: Co-morbidities (anxiety disorders, depression, psychosis), somatic disorders (hepatic function, malnutrition etc), timelines (AUD as lifetime diagnosis or at death, quantity of alcohol consumed, manner of death, cause of death, medication taken, etc etc.) It is critical to look at the clinical variables in depth as this is not a well-controlled animal study. In line 425 the authors refer to ED Fig 1a for cause and manner of death, which is not there. Smoking is the only comorbidity listed here and may be significantly different between groups (age, PMI as well?).

The number of DEGs is high for the sample size and I think the approach to use two different methods is good. However, can the authors provide QQ plots or other data that may support the notion that these results are not inflated? The inclusion of covariates in the MAST approach is basic. I understand that including a myriad of clinical covariates is difficult given the sample size, however, I wonder if a surrogate variable analysis could cover those known and unknown factors. Although the study here is the largest up to date on AUD, an independent replication would be critical due to the limit in sample size.

Given that the results presented are primarily associative, some functional manipulation of genes or cell types in animal

models would be critically important to better understand the generalizability of the data.

(Remarks on code availability)

Version 1:

Reviewer comments:

Reviewer #1

(Remarks to the Author)

The authors provided a very thorough and thoughtful revision. My questions have been addressed adequately. The one remaining detail is that the new mouse data is short-term forced administration and thus limits interpretation with human AUD. This caveat should be addressed in the Discussion.

(Remarks on code availability)

Response Letter for “Central amygdala single-nucleus atlas reveals chromatin and gene transcription dynamics in human alcohol use disorder”

Overview of our Response:

We thank each of the reviewers for their time, effort, and constructive feedback on our revision of the manuscript. Below, we respond to each of the reviewer’s specific comments and concerns in a point-by-point fashion. To summarize, we have conducted:

- Comparative analysis of different DEG models: We compared the DEG results for AUD vs CON on our snRNA-seq data using different methods: 1) our original MAST+Wilcox intersection; 2) pseudobulk approach with DESeq2 and 3) a MAST approach using additional clinical covariates (including additional psychiatric disorders, drug use, etc.). Similar GO terms were identified between DEGs from MAST+Wilcox and those from DESeq2. In addition, we found that >80% of DEGs overlapped between our original model and those including additional clinical covariates.
- Confirmation of Gene Expression Changes: We now include a series of fluorescent in situ (FISH) experiments for our top AUD GWAS DEGs. Small molecule FISH was used for *GRM8*, *CALN1*, *TCF4*, and *NCAM1* to confirm expression changes within INH subtypes.
- Additional Differential Gene Validation and Functional Analysis: As suggested, we have now added an animal model of alcohol use and performed the same molecular analyses as the human tissue including (snMultiome including RNA and ATAC) after mice were treated with a 2 g/kg injection of 100% EtOH.
- Expansion of sex-specific analyses: We have expanded our sex-specific analyses. We performed three sets of sex-specific DEG analyses and included them in new **Extended Data Fig. 4**. We now report Male AUD DEGs (**Supplementary Data Table 8**), Female AUD DEGs (**Supplementary Data Table 9**), Male vs Female Control differences (**Supplementary Data Table 10**) and expand on these findings in the text.
- Expansion of cell type analyses: We expanded our results and discussion to include more material about other cell types with genomic changes. We specifically look at astrocytes. Given the extensive number of DEGs and CREs we identified, we also highlighted expression changes, P2G linkages, and cell-type-specificity in 2 genes of astrocytes in **Extended Data Fig. 7g**:
- Changes to title and main text: As suggested, we have changed our current title to “*Central amygdala single-nucleus atlas reveals chromatin and gene transcription dynamics in human alcohol use disorder*”.
- Code update: We have uploaded our code on <https://github.com/mjgirgenti/AUDsnCEA>.

Reviewer #1 Comments

Ref 1.0 General Comments

Reviewer's comment	The manuscript by Lee et al., reports an interesting and important multi-omic single nucleus profiling of postmortem human amygdala from donors with alcohol use disorder (AUD). The study aims to identify cell-type specific dysregulated gene regulatory programs. The approach is novel, and the study could improve our understanding of the molecular underpinnings of AUD; however, the design, the analysis and the reporting contain significant shortcomings that should be addressed.
Author's response	We thank the reviewer for their review and have made a point-by-point response addressing their comments. Specifically, we have 1. Added new forms of experimental validations (ie. Mouse alcohol addiction model and fluorescent in situ of top DEGs) that validates many of our original conclusions.2. Added new DEGs analyses that account for sex, non-INH cell types, pseudobulk analysis, and new clinical covariates.3. Included new plots and tables clarifying our original results4. Shared our GitHub link and edited the manuscript Please see the detailed answers to your question below.

Ref 1.1 DEG Covariate

Reviewer's comment	Information on the metadata of the donors are significantly lacking, the authors do not report on the cause of death, lifetime consumption of alcohol, co-use of other substances of abuse (other than tobacco), presence or absence of psychiatric disorders, and related medications. All of which should have been included as covariates in the differential expression analyses.
Author's response	We thank the reviewer for the opportunity to clarify, and we agree that these covariates are important for our analysis. As suggested, we have added all the above metadata to new Supplementary Data Table 1. We have also included these in a new DEG analysis: 1. Included the new metadata in Supplementary Data Table 1

Sample	Alcohol Status	Lifetime Alcohol Consumption	AUD Severity	AUD Neurotox	Sex	Race	Age	PMI	RIN	Tob.ATOD	Meds.ATOD	Other Psychiatric Disorders	Illicit Drug Use	MOD
RT00256N	AUD	14.0	Severe	-	F	B	44	5.9	6.8	Y	Antidepressants	PTSD	2	Accidental
RT00256N	CON	0	-	-	F	W	60	25.0	6.6	N	Other medication(s)	-	0	Natural
RT00257N	CON	0	-	-	F	W	29	20.4	7.0	Y	Benzodiazepines Antidepressants Other medicati...	PTSD	1	Accidental
RT00258N	CON	0	-	-	F	W	39	12.8	9.0	N	Antidepressants	PTSD	0	Suicide
RT00260N	CON	0	-	-	F	W	65	21.5	9.1	N	Other medication(s)	PTSD	0	Natural

Supplementary Data Table 1. Samples Demographic Metadata

2. Performed a new DEG analysis that includes Manner of Death, Lifetime Alcohol Consumption, Illicit Drug Use, Other Psychiatric Disorders, and Meds ATOD as our new covariates. These new DEGs are included in **Supplementary Data Table 6**.

3. We found that there was 80% overlap with DEGs discovered in our original model (**Extended Data Fig. 4a**)

Extended Data Fig. 4a. Overlap of DEG analysis with new covariates.

Excerpt from the manuscript	To assess the impact of additional drug use and comorbidities, we performed additional analyses that included covariates for manner of death, lifetime alcohol consumption, illicit drug use, comorbid psychiatric disorders, and medications at time of death (Supplementary Data Table 6). Across cell types, ~80% of DEGs overlapped with our model (Extended Data Fig. 4a).
---

Ref 1.2 Validation

Reviewer's comment	A major limitation of the study is the lack of validation either in bulk tissues or alternative methods confirming cell specificity. Single cell transcriptome approaches are strengthened by including some experimental validations since many of the findings are computationally based.
Author's response	We thank the reviewer for this suggestion and agree that further experimental validations will strengthen these findings. As suggested. First, we performed fluorescent in situ hybridization (FISH). FISH allows for confirmation of expression changes within particularly cell

types and compartments. We chose to focus on the GWAS DEGs (**Fig. 3e**) as these seem to be most pertinent to AUD biology through their link with genetic risk. We designed probes for 4 transcripts: *GRM8*, *TCF4*, *CALN1*, and *NCAM1*. The probe designed for *GABRA2* failed, unfortunately. We were able to confirm expression changes for each of these genes. These are now included in **Extended Data Fig. 5c-j** and are also included below for the reviewers.

c, Example images of FISH from control (left) and AUD (right) tissue probed for *PENK* (red) and *NCAM1* (white). **d**, Average *NCAM1* pixel number per *PENK*+ nucleus. Mann-Whitney test, Control: n = 24 nuclei, AUD: n = 29 nuclei. **e**, Example images of FISH from control (left) and AUD (right) tissue probed for *PENK* (red) and *GRM8* (white). **f**, Average *GRM8* pixel number per *PENK*+ nucleus. Unpaired t test, Control: n = 10 nuclei, AUD: n = 9 nuclei. **g**, Example images of FISH from control (left) and AUD (right) tissue probed for *CALN1* (white). **h**, Average *CALN1* pixel number per nucleus. Unpaired t test, Control: n = 103 nuclei, AUD: n = 91 nuclei. **i**, Example images of FISH from control (left) and AUD (right) tissue probed for *TCF4* (white). **j**, Average *TCF4* pixel number per nucleus. Mann-Whitney test, Control: n = 189 nuclei, AUD: n = 119 nuclei. 3 control donors and 3 AUD donors were used for these analyses. 3 ROIs were analyzed per donor. *P* values indicated on violin plots. Green: Lipofusion. Blue: DAPI.

Second, we have also added an animal model of alcohol use and performed the same molecular analysis as the human tissue (snMultiome including RNA and ATAC) after mice were treated with an 2 g/kg injection of 100% EtOH.

1. *Mouse alcohol model validates CeA cell types:* INH, EXC, OLI, OPC, END, AST, MIC on snRNA, snATAC, and snMultiome embeddings, respectively (**Extended Data Fig. 9a-c**).

Extended Data Figure 9: Mouse snMultiome data. UMAP visualization across seven cell types of **a**, snRNA-seq **b**, snATAC-seq across seven cell types and **c**, snMultiome

2. *Mouse alcohol abuse model snMultiome-seq validates DEGs:* We identified 1814 total DEGs across all major cell types with most falling within INHa (**Extended Data Fig. 9d**). When compared with the human AUD DEG dataset we found overlap for 678 DEGs (**Extended Data Fig. 9e**).

Extended Data Figure 9: Mouse snMultiome data. **d**, Significant DEG counts in both directions for each major cell type. **e**, Intersection of alcohol DEGs between human and mouse. **f**, Top 25 enrichR GO terms of the 1,814 unique AUD DEG set.

3. *Mouse alcohol model snMultiome-seq to validates Chromatin Peaks:* We performed peak-to-gene linkage analysis on our mouse snMultiome data and identified 7485 total in the

mouse AUD model. Of these, 4710 overlapped with the human AUD CRE genes (62.9%) (**Extended Data Fig. 9h**).

Extended Data Figure 9: Mouse snMultiome data. h, Intersection of alcohol-focused Cis-Regulatory Element-linked genes between human and mouse.

4. *Mouse alcohol model snMultiome-seq to validate Gene Regulatory Networks.* We found that combined KLF16, KLF7 and KLF6 form a GRN that regulates 549 DEGs. We found 50 DEGs overlapped between the human and mouse GRN including DEG-GWAS gene *CALN1* confirming its role in AUD.

Extended Data Figure 9: Mouse snMultiome data. i, TF regulatory network showing the target genes for TFs *KLF6*, *KLF7*, and *KLF16* in INH. Peak-to-gene correlation > 0.9 was employed in visualizing the network.

Excerpt from
the manuscript

In order to functionally validate our genomic findings, we employed a mouse model of alcohol abuse. We isolated frozen amygdala from mice treated with an 2 g/kg injection of 100% EtOH. We performed single nuclei isolation and snMultiome in the same manner as our human donor cohorts. We identified all major cell types across all modalities: RNA (**Extended Data Fig. 9a**), ATAC (**Extended Data Fig. 9b**) and combined Multiome (**Extended Data Fig. 9c**) for 19,072 individual nuclei. We identified 1814 total DEGs across all major cell types with most falling in the INH (**Extended Data Fig. 9d**). When compared with the human AUD DEG dataset we found overlap for 678 DEGs (37.6%) (**Extended Data Fig. 9e**). Most DEGs overlapped in neuronal cell types and this was relected in gene set enrichment analysis which identified terms related to neuron projection, synaptic transmission, and glutamatergic signaling (**Extended Data Fig. 9f**). In addition, we intersected our INH DEGs with a previous central amygdala mouse model of alcohol withdrawal⁴⁶ and found 38 overlapping DEGs in INH and EXC neurons and MIC (**Extended Data Fig. 9g**). We performed peak-to-gene linkage analysis on our mouse snMultiome data and identified 7485 total in the mouse AUD model. Of these, 4710 overlapped with the human AUD CRE genes (62.9%) (**Extended Data Fig. 9h**). To confirm the function of our KLF gene regulatory network in human AUD (**Fig. 5g**), we constructed a GRN for these same TFs in our mouse model (**Extended Data Fig. 9i**). We found that combined KLF16, KLF 7 and KLF6 form a GRN that regulates 549 DEGs. We found 50 DEGs overlapped between the human and mouse GRN including DEG-GWAS gene *CALN1* confirming its likely role in alcohol abuse. Taken together, these data support the functional significance of KLF TFs in regulating calcium and glutamatergic signaling pathways identified in our dataset.

Ethanol Treatment (from **Methods**)

Mice were delivered at seven weeks of age and allowed two weeks to acclimate to the vivarium, where they were housed in a reverse light-dark cycle housing room. Two hours into the dark cycle, body weights were measured. Three hours into the dark cycle, mice received two sequential intraperitoneal injections each with a 10 ml/kg injection volume: the first was a solution of 5% DMSO in saline directly followed by 2 g/kg ethanol dissolved in water (n = 2M, 2F) or saline (n = 2M, 2F). Animals were randomly assigned to control or ethanol treated groups. Three hours following these

	injections, mice were euthanized by rapid decapitation and brains removed and immediately snap frozen on dry ice and then stored at -80°C. For tissue analyses, frozen brains were sliced into 2-mm coronal sections on a metal brain matrix (Plastics One, Roanoke VA) and amygdala samples were collected bilaterally with a 1-mm tissue punch (Fine Science Tools, Foster City, CA). Bilateral punches were collected from a coronal section targeted at -1.43 mm from bregma, according to76. Nuclei were isolated from rodent amygdala tissue as described in Brain Nuclei Isolation for Multiome methods section.
--	--

Ref 1.3 Protocol Clarify

Reviewer's comment	There is a discrepancy between the reported method and the nuclei isolation protocol, 10x Genomics Chromium Next GEM Single Cell Multiome ATAC + Gene Expression profiling which simultaneously profiling ATAC and RNA sequencing in the same nucleus and the reported nuclei isolation method implies producing two aliquots from each sample one of which is processed for single nucleus ATAC and one for gene expression profiling.
Author's response	We thank the Reviewer for the opportunity to clarify our isolation protocol. We have updated the method details in the manuscript. There was only one aliquot of nuclei split for 2 assays of snMultiome - RNA and ATAC.
Excerpt from the manuscript	Regions of interest were dissected on cryotome (leaflets of ~100-300 micrometers) from frozen Central Amygdala and stored at -80 °C. Cell nuclei isolation from brain sections were treated similarly to already established protocols with some modifications needed to have nuclei suitable for 10x Genomics Chromium Next GEM Single Cell Multiome ATAC + Gene Expression. To avoid experimental bias, nuclei isolation was done by the same person blinded for the metadata for the particular sample. All reagents were molecular biology grade and sourced from Sigma unless stated otherwise...

Ref 1.4 Cell Count 1

Reviewer's comment	The profiling yielded 183,391 cells from 50 samples (average 3600 nuclei per sample), the process as described targets 10,000 cells.
--

	Details should be provided whether the nuclei were eliminated during quality control, and report on quality control metrics for individual samples, or cells were lost during the library preparation process. How many cells were recovered from each sample?
Author's response	We thank the reviewer for the opportunity to clarify. Some cells are lost due to our rigorous QC. We have now included a new Supplementary Figure 3 showing the total number of nuclei (before and after QC) by sample. We outline the number of nuclei removed in each step below:  1. Starting with 618,861 cells, we removed the samples which had less than 2500 cells (612,963 cells). 2. We manually checked each sample on three different UMAPs, RNA-ATAC-Multiome specifically, and filtered for “nCount_ATAC > 1000, nCount_RNA > 200, and TSS.enrichment > 2” via Signac (533,337 cells). 3. We removed doublets – 63,268 cells were taken out with Scrublet, and 23,070 cells were taken out with DoubletDetection (446,999 cells). 4. After pooling all samples together, we filtered for cells that have “at most 10% mitochondrial genes, at least 200 genes, and at least 500 UMIs” via Pegasus (357,551 cells). 5. We then iteratively clustered on the RNA UMAP embedding for 10+ iterations and manually removed streaks of doublets that were not detected by Scrublet and DoubletDetection (233,591 cells). 6. Moving on the ATAC modality, we filtered for cells that had a transcription start site (TSS) enrichment score of at least four and at least ten mapped ATAC-seq fragments “filterTSS=4 and filterFrag=1000 in ArchR” (212,435 cells). 7. Finally, we removed five samples that most likely had “white matter” and other amygdala subregion contamination (174,188 cells). The recovered cell count for the final set of samples can be found in the new Supplementary Data Table 37 and Supplementary Data Fig. 3a. The excerpted 2 lines below are from the methods.

Excerpt from the manuscript	The final snRNA-seq data included 183,391 nuclei and 26,919 genes. We produced a final embedding of 174,188 high-quality nuclei in both RNA and ATAC modalities (Supplementary Data Figure 3, Supplementary Data Table 37).
---

Ref 1.5 Barcode Rank

Reviewer's comment	How many cells were taken out with Scrublet and DoubletDetection? ~183,000 cells is quite low for 55 samples. What does the barcode rank plot look like across all samples, authors denote high and low quality but don't show QC metrics for these samples.
Author's response	We thank the Reviewer for the opportunity to clarify:  63,268 cells were taken out with Scrublet, and 23,070 cells were taken out with DoubletDetection. While this number is lower than might be expected by our n, the number of nuclei

analyzed is consistent with and in some cases larger than other postmortem brain genomic studies

- The added barcode rank plots (before and after) demonstrate that only high-quality nuclei were retained (**Supplementary Data Fig. 3b**). Looking at the barcode rank plots, it contains only the high quality 'Cliff' portion and not the low quality 'Knee' background.

Supplementary Data Fig. 3b. Barcode rank plot across all samples, before and after data preprocessing.

- We determined high-quality samples from low-quality samples by inspecting all 55 UMAPs individually for each genomic modality, eliminating samples which did not cluster, resulting in a low cell count. **Fig. R1** has examples of samples which did not cluster.

Figure R1. High quality (top) and low quality (bottom) sample UMAPs across all modalities

Excerpt from the manuscript	We merged high-quality cells and high-quality samples in Pegasus (v1.5.0)³⁸, a python tool for analyzing transcriptomes of single cells, resulting in a total of 55 samples (Supplementary Data Figure 3). We produced a final embedding of 174,188 high-quality nuclei in both RNA and ATAC modalities (Supplementary Data Figure 3, Supplementary Data Table 37).
--

Ref 1.6 Method Clarify

Reviewer's comment	There is a discrepancy in reporting of the Wilcoxon test (de_analysis function from Pegasus in the methods and Seurat in the discussion section). The scripts used for performing MAST was not available at the time of review since link for the Github is not working. Please confirm active links to provide access to the scripts used in the study.
Author's response	We thank the Reviewer for bringing this to our attention.  • We have corrected the writing in the discussion because we used the Wilcoxon test from Pegasus throughout.

- Additionally, the Github link (<https://github.com/mjgirgenti/AUDsnCEA>) is now active (Fig. R2).

Figure R2. Screenshot of the working Github page.

- The Github link (https://github.com/mjgirgenti/AUDsnCEA/blob/main/scRNA/DEG_MAST.R) also has the MAST DEG script (Fig. R3).

Figure R3. Screenshot of MAST DEG script on the Github page.

Excerpt from the manuscript	To ensure that our findings are robust, we analyzed our data with both the Pegasus based Wilcoxon Test and the more conservative MAST method (including covariates).
--

Ref 1.7 DEG PseudoBulk

Reviewer's comment	MAST and wilcoxon rank sum testing are quite lenient differential methods in single cell studies. This is evident in the higher fold change and FDR cutoffs used in the study. Were these directly compared to pseudo bulk methods? Sample size can have unexpected impacts on most current DEG methods. Using two methods and then overlapping does help ensure that what DEGs are being identified are consistent but it would be good see pseudo bulking as well to determine the overlap between single nuclei based methods and a separate, more conservative approach for DE.																																
Author's response	We thank the reviewer for this suggestion. Pseudobulk DEGs are a more conservative approach and are becoming more popular in genomic studies. We were able to find additional DEGs with the pseudobulk DESeq2³⁹ (adjusted p-value < 0.05) method: 14 in inhibitory neurons, 26 in excitatory neurons, 1 in oligodendrocyte, 10 in astrocytes, and 1 in microglia.  • We show the overlap of pseudobulk and single-cell DEG lists below and they are included in Extended Data Fig. 4b.     Cell Type Single-cell only Overlap PB only     INH 751 18 14   EXC 541 5 26   OLI 38 1 0   OPC 35 0 0   END 448 1 0   AST 264 5 10   MIC 91 0 1    Extended Data Fig. 4b. Single-cell and PseudoBulk DEGs overlap  • We have also included the Pseudobulk DEG list in new Supplementary Data Table 7. 	Cell Type	Single-cell only	Overlap	PB only	INH	751	18	14	EXC	541	5	26	OLI	38	1	0	OPC	35	0	0	END	448	1	0	AST	264	5	10	MIC	91	0	1
Cell Type	Single-cell only	Overlap	PB only																														
INH	751	18	14																														
EXC	541	5	26																														
OLI	38	1	0																														
OPC	35	0	0																														
END	448	1	0																														
AST	264	5	10																														
MIC	91	0	1																														
Excerpt from the manuscript	Additionally, to ensure rigor, we tested a sample-based DEG method (i.e. pseudobulk) using DESeq2³⁹ (Supplementary Data Table 7). We identified additional DEGs: 14 in inhibitory neurons, 26 in excitatory neurons, 1 in oligodendrocyte, 10 in astrocytes, and 1 in microglia (Extended Data Fig. 4b).																																

Ref 1.8 Δ Celltype P-Value

Reviewer's comment	P values should be reported for proportional change showing no difference in cell types between AUD and Control.																																																																				
Author's response	We thank the reviewer for the opportunity to clarify, and we have since:  Added p-values into our paper as suggested.  (INH) 0.718; (EXC) 0.324; (OLI) 0.977; (OPC) 0.406; (END) 0.551; (AST) 0.945; and (MIC) 0.945. We include Fig. R4 for the reviewer to visually inspect the cell type proportion per cohort for each cell type.   <caption>Approximate data from Figure R4 box plot</caption>   Cell Type Cohort Median Proportion Q1 Q3     INH CON 0.12 0.08 0.16   AUD 0.12 0.08 0.18   EXC CON 0.04 0.02 0.06   AUD 0.04 0.02 0.06   OLI CON 0.44 0.18 0.54   AUD 0.33 0.23 0.55   OPC CON 0.10 0.06 0.14   AUD 0.10 0.06 0.12   END CON 0.02 0.01 0.03   AUD 0.02 0.01 0.03   AST CON 0.16 0.10 0.26   AUD 0.18 0.10 0.26   MIC CON 0.06 0.04 0.10   AUD 0.06 0.04 0.10     Figure R4. Cell count per sample for each cell type	Cell Type	Cohort	Median Proportion	Q1	Q3	INH	CON	0.12	0.08	0.16	AUD	0.12	0.08	0.18	EXC	CON	0.04	0.02	0.06	AUD	0.04	0.02	0.06	OLI	CON	0.44	0.18	0.54	AUD	0.33	0.23	0.55	OPC	CON	0.10	0.06	0.14	AUD	0.10	0.06	0.12	END	CON	0.02	0.01	0.03	AUD	0.02	0.01	0.03	AST	CON	0.16	0.10	0.26	AUD	0.18	0.10	0.26	MIC	CON	0.06	0.04	0.10	AUD	0.06	0.04	0.10
Cell Type	Cohort	Median Proportion	Q1	Q3																																																																	
INH	CON	0.12	0.08	0.16																																																																	
	AUD	0.12	0.08	0.18																																																																	
EXC	CON	0.04	0.02	0.06																																																																	
	AUD	0.04	0.02	0.06																																																																	
OLI	CON	0.44	0.18	0.54																																																																	
	AUD	0.33	0.23	0.55																																																																	
OPC	CON	0.10	0.06	0.14																																																																	
	AUD	0.10	0.06	0.12																																																																	
END	CON	0.02	0.01	0.03																																																																	
	AUD	0.02	0.01	0.03																																																																	
AST	CON	0.16	0.10	0.26																																																																	
	AUD	0.18	0.10	0.26																																																																	
MIC	CON	0.06	0.04	0.10																																																																	
	AUD	0.06	0.04	0.10																																																																	
Excerpt from the manuscript	We calculated cell type proportions based on the number of nuclei isolated (Fig. 1e) and found no significant proportion shifts due to disease (AUD versus CON): INH P-value=0.718; EXC P-value=0.324; OLI P-value=0.977; OPC P-value=0.406; END P-value=0.551; AST P-value=0.945; and MIC P-value=0.945.																																																																				

Ref 1.9 F13A1

Reviewer's comment	It is stated that microglia and macrophages were found but do not show markers to differentiate and collapse them further in the paper. F13a1 is not exclusive to macrophages and can be representative of monocyte-derived infiltrating macrophages and in some cases microglia. Cd44, CD169 and Siglech have been reported as good marker for infiltrating macrophages rather than resident microglia, in addition to higher expression of CD 9. What is the expression of a more robust microglia marker like TMEM119 or P2RY12? The identification of a true macrophage population is not possible without extensive markers and reporting the differences in key microglia homeostatic genes.
Author's response	We thank the reviewer for the suggestion. We have now checked our cell subtypes and integrated our original cell-subtyping with the reviewer's markers. F13A1 was established as a macrophage marker by Ma et al. in the 2022⁷⁷, and it emerged as a subpopulation marker de novo via the NSForest analysis in our data after we identified microglia using the established marker APBB1IP (Methods; Figure R6). The high co-expression of F13A1 and SIGLEC1 would suggest that this is a macrophage cluster.  • We found robust expression of microglia-specific markers P2RY12, TMEM119 (now included in Extended Data Fig. 1a). • We also found robust expression of infiltrating macrophage-specific marker SIGLEC1 (aka CD169) in our putative macrophage cluster that we previously identified with F13A1 and have included in Extended Data Fig. 1a. a 
Extended Data Fig. 1a. Microglia and Macrophage Gene Expression

- *CD44* and *CD9* have low expression in both clusters. (**Fig. R5**)

Figure R5. *CD44* and *CD9* gene expression in microglia and macrophage.

clusterName	f_score	PPV	TN	FP	FN	TP	marker_count	NSForest_markers	binary_genes
0	2	0.829596	0.827124	17382	116	106	555	1	['F13A1'] ['F13A1', 'COLEC12', 'CD163', 'MCTP1', 'STARD1...

Figure R6. Table of *de novo* identification of macrophage markers. The *f_score*, a harmonic mean of precision and recall, estimates the marker gene's importance. In this case, *F13A1* was able to delineate 82.96% of the macrophages from microglia.

Excerpt from the manuscript

We identified two vascular subtypes (endothelial cells, 39.8% and mural cells 60.2%) (**Fig 2e**); and two immune subtypes (microglia, 97.7% and macrophages, 2.3%) (**Fig 2f**). To confirm our immune cell subtypes, we examined expression of additional microglia-specific markers *P2RY12*, *TMEM119*, and found high co-expression with *APBB1IP*. Additionally, we examined the expression of infiltrating macrophage-specific marker *SIGLEC1* (aka *CD169*) and found high co-expression with our *F13A1* macrophage cluster (**Extended Data Fig. 1a**).

Ref 1.10 DEG Gender

Reviewer's comment

The authors report sex specific AUD changes and provide supplementary tables for male AUDvsCON and female AUDvsCON; however, there was no discussion of how the analysis were performed and specifics were not provided. Given the sample size this analysis is likely underpowered but potential sex differences should be examined and noted in the text.

Author's response

We thank the reviewer for this suggestion. We performed three sets of sex-specific DEG analyses and included them in new **Extended Data Fig. 4:**

1. Male AUD DEGs (**Supplementary Data Table 8**)
2. Female AUD DEGs (**Supplementary Data Table 9**)
3. Male vs Female Control differences (**Supplementary Data Table 10**)

Specifically, we intersected our original single-cell DEG list (**Methods**), with their respective sex exclusion/inclusion criteria. We have expanded our results section (excerpt below) and also added new **Extended Data Fig. 4c-f:**

- Significant DEG counts for males **Extended Data Fig. 4c**, in both directions for each major cell type (left) and INH sub cell types (right)

- Significant DEG counts for males **Extended Data Fig. 4d**, with their top enrichR Reactome (Pathway) terms.

- Significant DEG counts for females **Extended Data Fig. 4e**, in both directions for each major cell type (left) and INH sub cell types (right).

- Significant DEG counts for females **Extended Data Fig. 4f**, with their top enrichR Reactome (Pathway) terms.

Excerpt from the manuscript

Finally, we also observed sex-specific responses (**Supplementary Data Table 8-10**), consistent with previous observations in AUD. Specifically, males have the most DEGs in the INH cell type, whereas female DEGs are more evenly distributed across cell types (**Extended Data Figure 4c, e**). We also found pathway enrichment differences, specifically in WNT5A signaling in females and ERBB4 signaling in males (**Extended Data Fig. 4d, f**). In addition, male pathways tended to be more cell type-specific, while females' tended to be enriched across cell types. 710 DEGs were common across the male and female specific comparison or approximately 55% or 25% overlap, respectively.

Ref 1.11 CRE downDEGs

Reviewer's comment	Authors discuss the changes in chromatin accessibility that were linked into differentially expressed genes, these are potentially important findings, and additional details should be provided discussing instances where the changes of chromatin accessibility did not match observed gene expression changes and vice versa. Additional discussion of these findings is warranted.
Author's response	We thank the reviewer for the suggestion and have warranted additional discussion on this topic:  1. DEGs that did not have an associated CRE accounted for less than 10%. In terms of genes that did not have an associated CRE, only a few DEGs were discovered to not have an associated peak: 4.03% (INH), 2.38% (EXC), 10.5% (OLI), 0% (OPC), 9.35% (END), 8.18% (AST), 4.40% (MIC). 2. CREs that did not have an associated DEG can be explained by 3D genomic structure. There are more CRE peaks that do not have associated DEGs. Based on the 3-dimensional structure of the genome, many of the CREs that are not linked to DEGs likely extend to distal parts of the genome via chromatin-chromatin interactions. At present, we do not have Hi-C data for the amygdala to directly determine this. However, it stands to reason that around 30% of peaks are distal to DEGs, which makes long-range chromatin contacts the most likely scenario. 3. CREs with downregulated DEGs (63.95% of all CREs) – open chromatin region doesn't necessarily mean active transcription (Fig. R7). Of the DEG-linked CREs, there are 63.95% CREs associated with downregulated DEGs (Fig. R7). We posit several reasons for this:  a. the region may be accessible but not actively used, e.g., a poised enhancer or promoter that is open but lacks activating transcription factors; b. this could also indicate binding repressive transcription factors including silencers or repressive enhancers; c. An AUD downregulated gene might still have accessible regulatory regions due to feedback loops trying to restore expression.

	 Fig. R7. Intersection between open CREs and downregulated DEGs. Open chromatin region doesn't necessarily mean active transcription
Excerpt from the manuscript	To identify disease gene regulatory mechanisms, we intersected our CRE linked genes (CLGs) with DEGs for each cell type. 36.05% of CREs were linked to an upregulated DEG.

Ref 1.12 Astro EDFig7

Reviewer's comment	Additional detail and discussion about the changes in cell types other than inhibitory neurons should be provided and any overlaps between the identified differentially regulated elements in these different cell types. This could be presented in the form of a potential supplemental table.
Author's response	We thank the reviewer for this excellent suggestion. Given the importance of Astrocytes in AUD and the extensive number of DEGs and CREs we identified, we also highlighted expression changes, P2G linkages, and cell-type-specificity for 2 genes present in astrocytes (Extended Data Fig. 7g):  • NKAIN3 (logFC=-0.409 and FDR=2.06e-128) • FGFR3 (logFC=-1.298 and FDR=1.70e-96)   The figure is a violin plot labeled 'g' showing the expression of two genes, FGFR3 and NKAIN3, across seven cell types: INH, EXC, OLI, OPC, END, AST, and MIC. The y-axis represents expression levels from 0 to 5. For FGFR3, a significant upregulation (red asterisk) is shown in the AST cell type. For NKAIN3, a significant downregulation (blue asterisk) is shown in the AST cell type. Other cell types show no significant changes.  Extended Data Figure 7: Astrocytes multi-omic profile. g, Violin plot showing AUD versus CON expression of FGFR3, NKAIN3 in each cell type with significant up (red asterisk) and down (blue asterisk) expression.

We also added additional discussion of astrocytes to the manuscript.

- *NKAIN3*: UMAP plots of snRNA-seq gene expression on the **Extended Data Fig. 7a**, snRNA (left) and **Extended Data Fig. 7b**, snATAC-seq (right) embeddings. We see increased expression in astrocytes.

- *NKAIN3*: **Extended Data Fig. 7c**, Chromatin accessibility signal tracks highlighting *NKAIN3* peak-to-gene links across cell types. Differences across the seven cell types are indicated by the dashed boxes. We see increased accessibility in astrocytes.

- *FGFR3*: UMAP plots of snRNA-seq gene expression on the **Extended Data Fig. 7d**, snRNA (left) and **Extended Data Fig. 7e**, snATAC-seq (right) embeddings. We see increased expression in astrocytes.

- **FGFR3: Extended Data Fig. 7f**, Chromatin accessibility signal tracks highlighting *FGFR3* peak-to-gene links across cell types. Differences across the seven cell types are indicated by the dashed boxes. We see increased accessibility in astrocytes.

Excerpt from the manuscript

We also identified peak-to-gene links (strong CREs, correlation>0.8) for genes previously implicated in addiction in other cell types (e.g. astrocytes). For example, we find an extensive number of ATAC peak-to-gene links to *NKAIN3* and *FGFR3*. *NKAIN3* is a sodium/potassium pump gene that is highly expressed in astrocytes (**Extended Data Fig. 7a-c**) and changes in *FGFR3* transcript have been linked to alcohol exposure of offspring of binge drinking behavior⁴⁸ (**Extended Data Fig. 7d-f**); both TSS's are cell type-specific to astrocytes (**Extended Data Fig. 7c, f dashed box**). Additionally, both genes were significantly down regulated in AST (**Extended Data Fig. 7g**).

Ref 1.13 OPRM1

Reviewer's comment	The authors note a nominally significant up regulation of OPRM1 in INH neurons. This finding is somewhat surprising since OPRM1 is often not dysregulated in transcriptome studies specifically examining OUD. The donor metadata should be examined for comorbid opioid use or reported OUD/MMD medication usage that could potentially explain this interesting finding.
Author's response	We thank the reviewer for this feedback. We have removed this sentence due to there only being nominal upregulation. However, for the reviewers benefit, we have also verified that the nominal OPRM1 dysregulation is attributed to Alcohol Use Disorder and not comorbid opioid use:  There are six AUD cases with reported opioid use (colored red; Fig. R8). We found that OPRM1 expression does not differ among these six cases compared with all other AUD cases (P-value=0.23, non-significant).  Figure R8. OPRM1 Gene Expression by AUD-only Donor
Excerpt from the manuscript	Interestingly, we found a nominally significant up regulation of the mu opioid receptor (OPRM1) in INH neurons cell type and in both the DRD1 and LHX9 INH subtypes (Supplementary Data Table 2).

Ref 1.14 (minor) QC Clarify

Reviewer's comment	Line 96: (mean reads/nuclei), how it is written it suggests that this was the target sequencing depth.
Author's response	We have since changed the language to clarify the difference.
Excerpt from the manuscript	Here we present a single cell-type multi-omic analysis of 174,188 high-quality nuclei (average 6028 reads/nuclei) from postmortem human brain tissue of AUD and neurotypical controls (N =50).

Ref 1.15 (minor) Correct

Reviewer's comment	Figure 4b is not referenced in the text
Author's response	We thank the reviewer for pointing this out. Fig. 4b is now referenced in the second paragraph of the Results section "Cis-gene regulation of calcium activity and glutamatergic signaling genes is disrupted in AUD".
Excerpt from the manuscript	We then found 1,526,912 peak-to-gene links (correlation>0.45, FDR<1x10 ⁻⁴) using the union peaks and plotted side by side heatmaps of linked ATAC regions and gene expression (Fig. 4b).

Ref 1.16 (minor) Code

Reviewer's comment	Links to GitHub scripts were not working for this reviewer.
Author's response	We thank the reviewer for the feedback. The GitHub link should be working now: https://github.com/mjgirgenti/AUDsnCEA (Fig. R2).

Figure R2. Screenshot of the working Github page.

Excerpt from the manuscript

All code used in this study is freely available online and can be found at <https://github.com/mjgirgenti/AUDsnCEA>.

Reviewer #2 Comments

Ref 2.0 General Comments

Reviewer's comment	The manuscript by Lee et al provides insight into alcohol use disorder in humans by investigating single cell chromatin and gene expression data from post-mortem brain donors. The paper is well written and provides new data to the field. I'm not fully comfortable to provide an in-depth review of the bioinformatic analyses and therefore focus on clinical aspects primarily.
Author's response	We thank the reviewer for their constructive comments and have made a point-by-point response addressing them. Specifically, we have:  1. Clarified our writing in title, abstract, and introduction. 2. Incorporated a more comprehensive list of clinical covariates. 3. Added a functional validation using mouse amygdala in a mouse alcohol use model.

Ref 2.1 Title Clarify

Reviewer's comment	The title is slightly incorrect because the technique used is single nucleus multiome.
Author's response	We thank the reviewer's opportunity to clarify our sequencing protocol. We have since changed the title.
Excerpt from the manuscript	(Title) Central amygdala single-nucleus atlas reveals chromatin and gene transcription dynamics in human alcohol use disorder

Ref 2.2 Abstract Clarify

Reviewer's comment	I'd suggest including the sample size in the abstract.
Author's response	We thank the reviewer's suggestion and have since added the sample size in our abstract.
Excerpt from the manuscript	Here we present a multi-omic single nucleus study of ~175,000 nuclei from 50 donors with alcohol use disorder (AUD) and non-

	AUD controls, profiling cell type specific gene expression and chromatin accessibility in the human central amygdala.
--	---

Ref 2.3 Intro Clarify

Reviewer's comment	Line 89: A quick google search identifies snRNAseq studies in human CeA, I may be wrong.
Author's response	We thank the reviewer for the opportunity to clarify. While there are several existing snRNA-seq studies of the human amygdala,  1. There are currently none specific to Alcohol Use Disorder or to subregion-specific Central Amygdala in Humans. 2. However, we have toned down the language
Excerpt from the manuscript	To date there have been no single-cell multi-omic studies of the human CeA in AUD .

Ref 2.4 DSM-IV combine

Reviewer's comment	The clinical spectrum includes individuals with alcohol abuse and alcohol dependence, two very different phenotypes. What is the rationale and justification to combine both beyond the notion that DSM-V collapsed those two (can they be combined in this dataset given that the clinical heterogeneity within AUD is large)? The donor's age at death is young, what led to their demise? Do you have information on the severity of AUD (mild, moderate, severe) and the timeline across life? Did donors consume alcohol at death (neurotox data)? Do you have information on the neuropathological assessments of the donors (cases and controls), especially for those >60yrs [neurodegen panel]?
Author's response	We thank the reviewer for the opportunity to clarify this point of our study subjects.  • First, the reviewer is correct. In the latest DSM, alcohol abuse and dependence have been combined into AUD. From the point of view of alcohol exposure, there is quite a high degree of overlap between the level of drinking associated with alcohol abuse and alcohol dependence. Given the importance of these studies and the scarcity of human postmortem tissue and the current diagnostic criteria,

we felt that combining AUD cases with dependence and abuse was warranted. Each case is diagnosed by two psychiatrists and if there is disagreement in the diagnosis a third is consulted.

- Nevertheless, in an endeavor to alleviate the reviewers' concerns we tested whether there were any biological or statistical reasons to separate the cohorts. We calculated Pearson correlations between the transcriptomic profiles of all pairwise sample comparisons (**Fig. R9,10**). While we do find significant differences between AUD and CON samples, Alcohol Abuse and Dependence groups do not differ statistically.

Figure R9. Boxplot of Pearson correlations between each pair of samples, and **R10,** Boxplot of Pearson correlations between each pair of samples, after Log Normalization

- We also projected all samples using PCA for all phenotypic data (**Fig. R11**). We were able to discern clear distinctions between the AUDs (blue and green) and the CONs (orange). However the AUD subtypes (Dependence and Abuse) clustered together.

Figure R11. PCA of sample phenotypic metadata.

- Taken together, we feel that it is both biologically and statistically valid to combine Alcohol dependence and abuse samples into an AUD sample as diagnosed by the brain bank psychiatrists.
- We now include new meta data for:
 1. MOD (28 Natural, 15 Accidental, 6 Suicides, 1 Undetermined) in **Supplementary Data Table 1**.
 2. We have since included AUD Severity (6 Mild, 9 Moderate, and 7 Severe) in **Supplementary Data Table 1**.
 3. We have since included the 5 AUD samples that had AUD Neurotox in addition to many other covariates in **Supplementary Data Table 1**.

Sample	Alcohol Status	Lifetime Alcohol Consumption	AUD Severity	AUD Neurotox	Sex	Race	Age	PMI	RIN	Tob.ATOD	Meds.ATOD	Other Psychiatric Disorders	Illicit Drug Use	MOD
RT00255N	AUD	14.0	Severe	-	F	B	44	5.9	6.8	Y	Antidepressants	PTSD	2	Accidental
RT00256N	CON	0	-	-	F	W	60	25.0	6.6	N	Other medication(s)	-	0	Natural
RT00257N	CON	0	-	-	F	W	29	20.4	7.0	Y	Benzodiazepines Antidepressants Other medicati...	PTSD	1	Accidental
RT00258N	CON	0	-	-	F	W	39	12.8	9.0	N	Antidepressants	PTSD	0	Suicide
RT00260N	CON	0	-	-	F	W	65	21.5	9.1	N	Other medication(s)	PTSD	0	Natural

Supplementary Data Table 1. Samples Demographic Metadata

- Finally, all donated brains underwent physical examination by a board-certified neuropathologist at both the macroscopic and microscopic level, please see our excerpt below on how these examinations are conducted.

Excerpt from the manuscript	All brain tissue underwent physical examination by a board-certified neuropathologist at both the macroscopic and microscopic level. All tissue was screened for confounding neuropathologies, including amyloid plaques, Lewy bodies (Parkinson’s disease and dementia with Lewy bodies), and transactive response DNA binding protein-43 (TDP-43) (ALS, frontotemporal dementia, and limbic predominant age-related TDP-43 encephalopathy). All cases were also evaluated at a gross (macroscopic) level to detect evidence of atrophy, trauma, or infarction. Importantly, cases were excluded if massive trauma occurred (from head injury) that severely damaged the tissue, if the tissue was damaged from a stroke involving a large portion of the brain, if the decedent was on a respirator for an extended period of time, if the donor had brain cancer, or if the donor had a history of HIV, AIDS, COVID-19, or other communicable disease.
--

Ref 2.5 Look Covariate

Reviewer’s comment	Looking at the Extended Data Fig 1a, the details given regarding the post-mortem samples is very limited. The U Pitt Brain Bank has a large amount of data on their donors that is critical to include and to evaluate the data and this manuscript. For example: Co-morbidities (anxiety disorders, depression, psychosis), somatic disorders (hepatic function, malnutrition etc), timelines (AUD as lifetime diagnosis or at death, quantity of alcohol consumed, manner of death, cause of death, medication taken, etc etc.) It is critical to look at the clinical variables in depth as this is not a well-controlled animal study. In line 425 the authors refer to ED Fig 1a for cause and manner of death, which is not there. Smoking is the only comorbidity listed here and may be significantly different between groups (age, PMI as well?).
Author’s response	We thank the reviewer for the opportunity to explain more about the donors. We have now included additional details (Supplementary Data Table 1) including:  ● Other Psychiatric Disorders ● Lifetime Alcohol Consumption (Range 0~46) ● AUD Severity (6 Mild, 9 Moderate, and 7 Severe) ● AUD Neurotoxicity ATOD (for 5 AUD samples)

- MOD (28 Natural, 15 Accidental, 6 Suicides, 1 Undetermined)
- Medications ATOD
- Illicit Drug Use (Range 0~4)

in addition to our original covariates Sex, Race, Age, PMI, RIN.

We did find that while 80% of DEGs overlap (**Extended Data Fig. 4a**) they are nevertheless significantly different datasets. The alcohol attributes (e.g. Lifetime Alcohol consumption), Smoking, Medications, Other Psych, and Illicit Drug Use were significantly different between groups. Therefore, we have included a new DEG list to reflect this (**Supplementary Data Table 6**).

Extended Data Fig. 4a. Overlap of DEG analysis with new covariates.

Sample	Alcohol Status	Lifetime Alcohol Consumption	AUD Severity	AUD Neurotox	Sex	Race	Age	PMI	RIN	Tob.ATOD	Meds.ATOD	Other Psychiatric Disorders	Illicit Drug Use	MOD
RT00255N	AUD	14.0	Severe	-	F	B	44	5.9	6.8	Y	Antidepressants	PTSD	2	Accidental
RT00256N	CON	0	-	-	F	W	60	25.0	6.6	N	Other medication(s)	-	0	Natural
RT00257N	CON	0	-	-	F	W	29	20.4	7.0	Y	Benzodiazepines Antidepressants Other medicati...	PTSD	1	Accidental
RT00258N	CON	0	-	-	F	W	39	12.8	9.0	N	Antidepressants	PTSD	0	Suicide
RT00260N	CON	0	-	-	F	W	65	21.5	9.1	N	Other medication(s)	PTSD	0	Natural

Supplementary Data Table 1. Samples Demographic Metadata

Excerpt from the manuscript

Sociodemographic and clinical details are listed in **Supplementary Data Table 1** and include the manner of death, alcohol-related phenotypes, tobacco use, additional psychiatric disorders, and illicit drug use.

Ref 2.6 Inflated DEGs

Reviewer's comment

The number of DEGs is high for the sample size and I think the approach to use two different methods is good. However, can the authors provide QQ plots or other data that may support the notion that these results are not inflated? The inclusion of covariates in the MAST approach is basic. I understand that including a myriad of clinical covariates is difficult given the sample size, however, I wonder if a surrogate variable analysis could cover those known and unknown factors. Although the study here is the largest up to

date on AUD, an independent replication would be critical due to the limit in sample size.

Author's response

We agree with the reviewer that multiple models should be employed in the detection of DEGs. Since our last submission, we have provided the four additional models. Taken together, our results show that our DEG analysis is likely not inflated:

1. *New DEGs which include additional clinical covariates and pseudobulk analysis (Supplementary Data Table 6, 7).* We have found that, with the inclusion of new clinical variates (more conservative model) overlapped ~80% with previously found DEGs (**Extended Data Fig. 4a**).

Extended Data Fig. 4a. Overlap of DEG analysis with new covariates.

2. *Batch-corrected DEGs via surrogate variable analysis.* We have performed surrogate variable analysis via “Harmony”. The data now reflects batch-corrected DEGs to look for hidden batch effects (**Fig. R12**).

Fig. R12. UMAP Plots before and after surrogate variable analysis

3. *Validation using small molecule FISH of GWAS DEGs.* Please see R1.4 Validation on page 3.
4. *Replication and validation using a mouse model of alcohol use.* We identified 1814 total DEGs across all major cell types with most falling in the INH (**Extended Data Fig. 9d**). When compared with the human AUD DEG dataset we found a very high overlap of 678 DEGs (~40%) (**Extended Data Fig. 9e**).

Extended Data Figure 9: Mouse snMultiome data. **d**, Significant DEG counts in both directions for each major cell type. **e**, Intersection of alcohol-focused DEGs between human and mouse.

5. *QQ Plot to visualize the distribution of DEG p-values.* We also visualize our Wilcoxon-intersected MAST DEGs with a QQ Plot (**Fig. R13**). These results seem minorly inflated, but this is a generally accepted consequence of single-cell differential expressed genes. Other methods which dealt with single-cell gene expression have also shown similar results.

Figure R13. QQ Plot of Wilcoxon-intersected MAST DEG p-values.

Excerpt from the manuscript

Ethanol Treatment (from Methods)
Mice were delivered at seven weeks of age and allowed two weeks to acclimate to the vivarium, where they were housed in a reverse light-dark cycle housing room. Two hours into the dark cycle, body weights were measured. Three hours into the dark cycle, mice received two sequential intraperitoneal injections each with a 10 ml/kg injection volume: the first was a solution of 5% DMSO in saline directly followed by 2 g/kg ethanol dissolved in water (n = 2M, 2F) or saline (n = 2M, 2F). Animals were randomly assigned to control or ethanol treated groups. Three hours following these injections, mice were euthanized by rapid decapitation and brains removed and immediately snap frozen on dry ice and then stored at -80°C. For tissue analyses, frozen brains were sliced into 2-mm coronal sections on a metal brain matrix (Plastics One, Roanoke VA) and amygdala samples were collected bilaterally with a 1-mm tissue punch (Fine Science Tools, Foster City, CA). Bilateral punches were collected from a coronal section targeted at -1.43 mm from bregma, according to⁷⁶. Nuclei were isolated from rodent amygdala tissue as described in Brain Nuclei Isolation for Multiome methods section.

Reviewer's comment	Given that the results presented are primarily associative, some functional manipulation of genes or cell types in animal models would be critically important to better understand the generalizability of the data.
Author's response	We thank the reviewer for this suggestion and agree that further experimental validations will strengthen these findings. As suggested, we have now added an animal model of alcohol use and performed the same molecular analyses as the human tissue including (snMultiome including RNA and ATAC) after mice were treated with an 2 g/kg injection of 100% EtOH.  1. Mouse alcohol model validates CeA cell types: INH, EXC, OLI, OPC, END, AST, MIC on snRNA, snATAC, and snMultiome embeddings, respectively (Extended Data Fig. 9a-c).  Figure 9 consists of three UMAP plots labeled a, b, and c. Plot a is titled 'RNA' and shows clusters for MIC, OLI, INH, EXC, OPC, END, and AST. Plot b is titled 'ATAC' and shows clusters for MIC, EXC, INH, END, OPC, and AST. Plot c is titled 'Multiome' and shows clusters for AST, INH, END, OLI, EXC, OPC, and MIC. Each plot has UMAP1 on the x-axis and UMAP2 on the y-axis.  Extended Data Figure 9: Mouse snMultiome data. UMAP visualization across seven cell types of a, snRNA-seq b, snATAC-seq across seven cell types and c, snMultiome  2. Mouse alcohol model snMultiome-seq validates DEGs: We identified 1814 total DEGs across all major cell types with most falling in the INH (Extended Data Fig. 9d). When compared with the human AUD DEG dataset we found overlap for 678 DEGs (~40%) (Extended Data Fig. 9e).

Extended Data Figure 9: Mouse snMultiome data. d, Significant DEG counts in both directions for each major cell type. **e**, Intersection of alcohol-focused DEGs between human and mouse. **f**, Top 25 enrichR GO terms of the 1,814 unique AUD DEG set.

3. *Mouse alcohol model snMultiome-seq validates Chromatin Peaks:* We performed peak-to-gene linkage analysis on our mouse snMultiome data and identified 7485 total in the mouse AUD model. Of these, 4710 overlapped with the human AUD CRE genes (62.9%) (**Extended Data Fig. 9h**).

Extended Data Figure 9: Mouse snMultiome data. h, Intersection of alcohol-focused *Cis*-Regulatory Element-linked genes between human and mouse.

4. *Mouse alcohol model snMultiome-seq validates AUD GRN.* We found that combined KLF16, KLF 7 and KLF6 form a GRN that regulates 549 DEGs (**Extended Data Fig. 9i**). We found 50 DEGs overlapped between the human and mouse GRN including DEG-GWAS gene *CALN1* confirming its likely role in alcohol abuse.

Extended Data Figure 9: Mouse snMultiome data. i, TF regulatory network showing the target genes for TFs *KLF6*, *KLF7*, and *KLF16* in INH. Peak-to-gene correlation > 0.9 was employed in visualizing the network.

Excerpt from the manuscript

In order to functionally validate our genomic findings, we employed a mouse model of alcohol abuse. We isolated frozen amygdala from mice treated with an 2 g/kg injection of 100% EtOH. We performed single nuclei isolation and snMultiome in the same manner as our human donor cohorts. We identified all major cell types across all modalities: RNA (Extended Data Fig. 9a), ATAC (Extended Data Fig. 9b) and combined Multiome (Extended Data Fig. 9c) for 19,072 individual nuclei. We identified 1814 total DEGs across all major cell types with most falling in the INH (Extended Data Fig. 9d). When compared with the human AUD DEG dataset we found overlap for 678 DEGs (37.6%) (Extended Data Fig. 9e). Most DEGs overlapped in neuronal cell types and this was reflected in gene set enrichment analysis which identified terms related to neuron projection, synaptic transmission, and glutamatergic signaling (Extended Data Fig. 9f). In addition, we intersected our INH DEGs with a previous central amygdala mouse model of alcohol withdrawal⁴⁶ and found 38 overlapping DEGs in INH and EXC neurons and MIC (Extended Data Fig. 9g). We performed peak-to-gene linkage analysis on our mouse snMultiome data and identified 7485 total in the mouse AUD model. Of these, 4710 overlapped with the human AUD CRE genes (62.9%) (Extended Data Fig. 9h). To confirm the function of our KLF

	gene regulatory network in human AUD (Fig. 5g), we constructed a GRN for these same TFs in our mouse model (Extended Data Fig. 9i). We found that combined KLF16, KLF 7 and KLF6 form a GRN that regulates 549 DEGs. We found 50 DEGs overlapped between the human and mouse GRN including DEG-GWAS gene CALN1 confirming its likely role in alcohol abuse. Taken together, these data support the functional significance of KLF TFs in regulating calcium and glutamatergic signaling pathways identified in our dataset.
--	---

Response Letter for “Central amygdala single-nucleus atlas reveals chromatin and gene transcription dynamics in human alcohol use disorder”

Overview of our Response:

We thank each of the reviewers for their time, effort, and constructive feedback on our revision of the manuscript. Below, we respond to each of the reviewer’s specific comments and concerns in a point-by-point fashion. To summarize, we have conducted:

Reviewer #1 (Remarks to the Author):

The authors provided a very thorough and thoughtful revision. My questions have been addressed adequately. The one remaining detail is that the new mouse data is short-term forced administration and thus limits interpretation with human AUD. This caveat should be addressed in the Discussion.

We thank the reviewer for this suggestion. We have added the following text to the Discussion:

*We were also able to confirm the presence of a GRN of KLF6, 7 and 16 in our animal model of alcohol use (**Extended Data Figure 9**). However, this was a short-term administration of ethanol and thus limits some of the interpretation of this finding.*